# Influence of fast ice on future ice shelf melting in the Totten Glacier area, East Antarctica

Guillian Van Achter[1], Thierry Fichefet[1], Hugues Goosse[1], and Eduardo Moreno-Chamarro[2]

[1]Earth and Life Institute, Georges Lemaitre Centre for Earth and Climate Research, UCLouvain, Louvain-la-Neuve, Belgium.
[2]Barcelona Supercomputing Center (BSC), Barcelona, 08034, Spain.

**Correspondence:** Guillian Van Achter (guillian.vanachter@uclouvain.be)

**Abstract.** The Totten Glacier in East Antarctica is of major climatic interest because of the large fluctuations of its grounding line and potential vulnerability to climate change. Here, we use a series of high-resolution, regional NEMO-LIM-based experiments, which include an explicit treatment of ocean–ice shelf interactions as well as a representation of grounded icebergs and fast ice, to investigate the changes in ocean–ice interactions in the Totten Glacier area between the recent past (1995-2014) and the end of the 21st century (2081-2100) under SSP4-4.5 climate change conditions. By the end of the 21st century, the wide areas of multiyear fast ice simulated in the recent past are replaced by small patches of first year fast ice along the coast, which decreases the total summer sea ice extent. The Antarctic Slope Current is accelerated by about 116%, which decreases the heat exchange across the shelf and tends to reduce the ice shelf basal melt rate, but this effect is counterbalanced by the effect of the oceanic warming. As a consequence, despite the accelerated Antarctic Slope Current, the Totten ice shelf melt rate is increased by 91% due to the intrusion of warmer water into its cavity. The representation of fast ice dampens the ice shelf melt rate increase throughout the 21st century, as the Totten ice shelf melt rate increase reaches 136% when fast ice is not taken into account. The Moscow University ice shelf melt rate increase is even more impacted by the representation of fast ice, with a 36% melt rate increase with fast ice, compared to a 75% increase without a fast ice representation. This influence of the representation of fast ice in our simulations on the basal melting rate trend over the 21st century is explained by the large impact of the fast ice for present-day conditions ($\sim 25\%$ difference in m/yr), while the impact decreases significantly in the end of the 21st century ($\sim 4\%$ difference in m/yr). As a consequence, the reduction of the fast ice extent in the future induces a decrease of the fast ice effect on the ice shelf melt rate that partly compensates for the increase due to warming of the ocean. This highlights the importance of including a representation of fast ice to simulate realistic ice shelf melt rate increase in East Antarctica under warming conditions.

## 1 Introduction

The Totten Glacier area, located on the Sabrina Coast in East Antarctica, underwent significant grounding-line fluctuations during the recent past. Driven by changes in the ocean (Aitken et al., 2016), these fluctuations are making the region potentially vulnerable to rapid ice sheet collapse (Roberts et al., 2011). There has been some indication of ice shelf thinning during the last decade (Khazendar et al., 2013), although it remains unclear whether this represents a long-term trend (Paolo et al.,

2015). Furthermore, the Totten catchment, located in the Aurora Subglacial Basin of East Antarctica, contains 3.5-m sea level rise equivalent and is one of the few sectors of East Antarctica where changes in ice dynamics have been observed recently (Greenbaum et al., 2015). Understanding how changes in the ocean–ice interactions are interfering with the basal melt of the Antarctic ice shelves and how they will evolve in the future is crucial for projections of future sea level rise.

A key element of the ocean–ice interactions in the Totten Glacier area is the fast ice (Van Achter et al., 2022), defined as
stationary sea ice which forms and remains attached to the shore or between grounded icebergs (WMO, 1970; Massom et al., 2001; Fraser et al., 2012). Numerous observations show the presence of both multiyear and seasonal fast ice in front of both the Totten and Moscow University ice shelves (Fraser et al., 2012, 2020). Van Achter et al. (2022) have demonstrated with a numerical model (over the years 2001-2010) that the presence of fast ice in the Totten Glacier region impacts the whole ice–ocean system. In this region, the ocean surface covered by fast ice mainly increases through the advection of sea ice which forms
ice arches between icebergs or between icebergs and the coast. Once sea ice is trapped by the ice arches, it thickens by snow accumulation and subsequent snow ice formation. Once established, a thick multiyear fast ice pack thermodynamically isolates the ocean from the atmosphere during summer. In winter, both yearly and multiyear fast ice relocate the coastal polynyas off shore, which decreases the sea ice production close to the coast. These effects both increase the ocean stratification in front of the cavities and favour the intrusion of modified Circumpolar Deep Water (mCDW) into the ice shelf cavities, with an enhanced
ice shelf basal melting. Fast ice can also have a dynamical influence on the ice shelf, as the loss of buttressing from the break-up of seasonal fast ice increases the seasonality of the Totten ice shelf (TIS) basal melt rate close to the ice front (Greene et al., 2018).

Large density, temperature, salinity and sea level gradients are found across the Antarctic Slope Front (ASF; Whitworth et al., 1985; Jacobs, 1991), which separates the continental shelf from the open Southern Ocean. A strong pressure gradient is
observed across the ASF, mainly caused by the strong easterly winds that drive a sea surface height gradient via Ekman drift (Mathiot et al., 2011), as well as a density gradient, which results from the differences in temperature and salinity of the water masses across the ASF. Additionally, the ASF manifests itself through strong isopycnal doming towards the continental shelf. These lateral gradients across the ASF contribute to establishing the geostrophically balanced, vertically sheared along-slope flows of the Antarctic Slope Current (ASC; Jacobs, 1991; Thompson et al., 2018). The ocean dynamics associated with the
ASF and ASC govern along- and across-slope heat transport (Stewart et al., 2018), and act as a barrier to mixing between shelf and open-ocean waters (Thompson et al., 2018). Shifts in position of the ASF, or changes in the range of densities of waters that occupy the continental shelf, therefore strongly influence the heat budget of the continental shelf (Thompson et al., 2018). Moorman et al. (2020) suggested that increasing glacial meltwater fluxes strengthens the lateral density gradient associated with the ASF, which reduces cross-slope water exchanges and isolates shelf waters from warm mCDW. Naughten et al. (2018)
also found an intensified density gradient across the continental slope which reinforces the Antarctic Coastal Current. In the Totten Glacier region, the ASC modulates the heat intrusion towards the Totten Glacier (Nakayama et al., 2021).

As a consequence, understanding how the ASC will evolve in this region under future climate conditions is key to gain insights on changes in heat intrusion across the continental shelf break. The future changes in ice shelf melt rate under different Representative Concentration Pathway (RCP) scenarios have been studied with both global and regional models (Hellmer et al.,

2012; Timmermann and Goeller, 2017). In the Totten Glacier area, Pelle et al. (2021) found that, by the end of the 21st century, the ASC might weaken by 37% compared to its present-day state and the Totten ice shelf melt rate might increase by 56% following a high emission scenario. Those models include representations of ocean–ice shelf interactions, but none of them has an prognostic representation of the fast ice.

The present study follows on from Van Achter et al. (2022), which presented a prognostic fast ice representation and investigated the impact of fast ice on ocean–ice interactions over the last decade. The goal of the present study is twofold. As future climate warming will potentially lead to a decrease of the sea ice cover in the Southern Ocean and hence most likely to a reduction in the stability and duration of the fast ice cover that will affect atmosphere-ocean fluxes, oceanic stratification and ocean currents, we first evaluate how the ocean–ice shelf interactions in the Totten Glacier region will change in a warming climate, with a particular focus on the ASC changes. Secondly, we aim at assessing how an explicit fast ice representation included in a model affects the simulation of the ice shelf melt rate evolution between the recent past and the end of the 21st century. In order to answer these questions, we designed six simulations with a high-resolution, regional configuration of the NEMO3.6-LIM3 model, four of them being forced with anomalies derived from a simulation with the global climate model EC-Earth3 driven by the SSP4-4.5 scenario (Shared Socioeconomic Pathways; Döscher et al., 2021).

This manuscript is organised as follows. The model, regional configuration and experimental design are described in Section 2. In Section 3, we analyse the changes in sea ice and ocean characteristics and ice shelf melt rate between the recent past and the end of the 21st century simulated by the model. The sensitivity of the ice shelf melt rate to the representation of fast ice is then addressed in Section 4. Conclusions are finally given in Section 5.

## 2  The model, forcing and experimental design

### 2.1  Ocean–sea ice model

We make use of NEMO 3.6 (Nucleus for European Modelling of the Ocean; Madec, 2008) that includes the ocean model OPA (océan parallélisé) coupled with the Louvain-la-Neuve sea ice model (LIM3; Vancoppenolle et al., 2009; Rousset et al., 2015). This combination is hereafter referred to as NEMO-LIM. OPA is a state-of-the-art, finite-difference ocean model based on primitive equations. Our setting includes a polynomial approximation of the seawater equation of state (TEOS-10, IOC, 2010) optimized for a Boussinesq fluid (Roquet et al., 2014). Vertical turbulent mixing is rendered through a Turbulent Kinetic Energy (TKE) scheme (Bougeault and Lacarrere, 1989; Gaspar et al., 1990; Madec et al., 1998). The enhanced vertical diffusion mixing coefficient utilised in this scheme is fixed to 20 m$^2$/s. LIM3 uses a five-category subgrid-scale distribution of sea ice thickness (Bitz et al., 2001). The drag coefficient is set to $7.1 \times 10^{-3}$ at the sea ice–ocean interface and $2 \times 10^{-3}$ at the sea ice–atmosphere one (Massonnet et al., 2014). Ice shelf cavities with explicit ocean—ice shelf interactions are represented by the ice shelf module implemented in NEMO by Mathiot et al. (2017), using the three-equation formulation from Jenkins (1991). Transfer coefficients for heat ($\gamma_T$) and salt ($\gamma_S$) between the ocean and ice shelves are velocity dependent (Dansereau et al., 2014): $\gamma_{T,S} = \Gamma_{T,S} \times u_*$. The friction velocity is given by $u_* = C_d \times \sqrt{u_{TML}^2}$ and constant values of $\Gamma_T$ and $\Gamma_S$ taken from Jourdain et al. (2017) are employed ($\Gamma_T = 2.21 \times 10^{-2}$ and $\Gamma_S = 6.19 \times 10^{-4}$ for temperature and salinity, respectively).

$C_d$ is the top drag coefficient, set to $3 \times 10^{-3}$, and $u_{TML}$ is the ocean velocity in the top mixed layer, which is either the top 30 m of the water column or the top model layer (if thicker than 30 m) (Losch, 2008).


## 2.2 The Totten24 model configuration

Here, we use a regional configuration of NEMO-LIM, referred to as Totten24, which is described in detail in Van Achter et al. (2022). The horizontal grid is a 1/24°refinement (less than 2 km grid spacing) of the eORCA1 tripolar grid, centered on the continental shelf in front of the TIS, East Antarctica, and covering an area between 108-129° E and 63-68° S (Fig. 1). The

NEMO and LIM time steps are 150 s and 900 s, respectively. The vertical discretisation has 75 levels, with level thickness increasing with depth and partial cells used for better representing bedrock and ice shelf bases (Adcroft et al., 1997). The ocean layer directly underneath the ice shelf base varies between 30 m near the cavity front and 80 m in the center of the cavity. The bathymetry and ice shelf draft datasets are derived from the NASA Making Earth System Data Records for Use in Research Environments (MEaSUREs) program, which contains a bathymetry map of Antarctica based on mass conservation, streamline

diffusion and other methods (Morlighem et al., 2020).

The ocean lateral boundary conditions and initial conditions are taken from a 1979-2014 simulation with an eORCA025 (1/4°, 75 levels) peri-Antarctic NEMO-LIM configuration (Pelletier et al., 2022) (hereafter referred to as PARASO). Because of a negative salinity bias in the PARASO simulation, a salinity correction of 0.25 g/kg is uniformly added to the ocean lateral boundary conditions and initial conditions. At the lateral boundaries, a flow relaxation scheme (Engedahl, 1995) is applied

to the three-dimensional ocean variables and two-dimensional sea ice variables. A Flather scheme (Flather, 1994) is used for barotropic velocities and sea surface elevation. Furthermore, the sea surface elevation and barotropic velocities from the FES2014 tide model (Carrère et al., 2012) are added to the boundary for the tide components K1, K2, M2, P1, O1, S2, 2N2, Mm, M4, Mf, Mtm, MU2, N2, NU2, Q1, S1, L2, T2, as in Maraldi et al. (2013); Jourdain et al. (2019); Huot et al. (2021). The surface fluxes of heat, freshwater and momentum are computed using the CORE bulk formulas (Large and Yeager, 2004),

with atmosphere input coming from the fifth generation ECMWF atmospheric reanalysis (ERA5, Hersbach et al., 2020). No surface salinity restoring is applied.

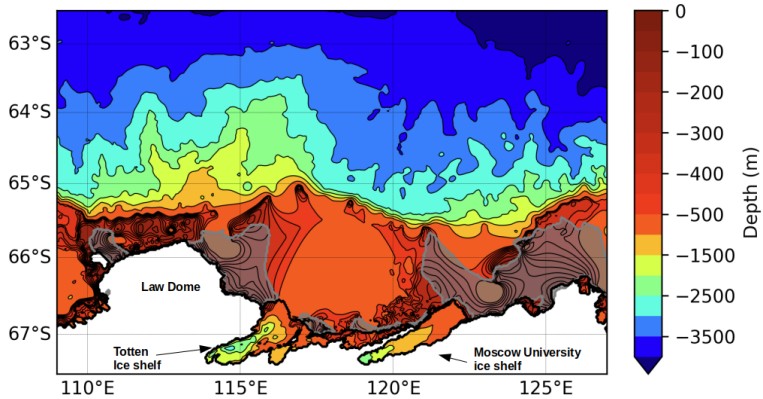

**Figure 1.** Model bathymetry and domain. The contour interval is 50 m up to 500 m depth and 500 m up to 4500 m depth. Ice shelf cavities are surrounded by a thick black line. The 0.75 fast ice observed frequency from Fraser et al. (2020) is shown by the shaded gray areas.

## 2.3 Experimental design

Our experimental design consists of one reference simulation and a set of five sensitivity experiments. All simulations include the tide constituents and the ocean–ice shelf interactions (i.e., open ice shelf cavities and interactive basal melt computation).

The reference simulation (REF) includes a representation of grounded icebergs and a sea ice tensile strength parameterisation. Both are needed to simulate adequately the fast ice formation (Van Achter et al., 2022). The grounded iceberg dataset used is extracted from the remote sensed mosaic 'RAMP AMM-1 SAR Image Mosaic of Antarctica, Version2' (Jezek et al., 2013) and covers the September-October months of 1997. The grounded icebergs are prescribed in the model by setting the bathymetry value to zero at every iceberg location (Van Achter et al., 2022). The sea ice tensile strength parameterisation was developed

by Lemieux et al. (2016). The REF simulation covers the 1995 to 2014 period, with a 20-yr spin-up (the 20-yr simulations are ran twice). A similar simulation was conducted by Van Achter et al. (2022) and evaluated against observations (sea ice concentration, fast ice, sea ice production, sea ice thickness, polynya locations and temperature and salinity distributions). For the present study, the salinity bias identified in Van Achter et al. (2022) has been corrected, without altering the vertical profiles of temperature (Fig. 2a), by adding 0.2 $g/kg$ in salinity to the oceanic lateral boundary conditions (Fig. 2b). Moreover, due to

a miscalculation in Van Achter et al. (2022) in the computation of the temporal basal melt rate, the top drag coefficient in the ice shelf cavities has been decreased from $8 \times 10^{-3}$ to $3 \times 10^{-3}$. With these modifications, the simulated TIS melt rate (11.13 m/yr) is in better agreement with Rignot et al. (2013)'s estimate (10.47 $\pm$ 0.7 m/yr). The Moscow University ice shelf (MUIS) basal melt rate is 7.73 $\pm$ 2.51 m/yr, which slightly overestimates the 4.7 $\pm$ 0.8 m/yr Rignot et al. (2013)'s estimate. Except for those changes in ice shelf melt rate and salinity profiles, results from this new REF simulation are very similar to those of the

previous one in terms of sea ice distribution and ocean circulation.

The sensitivity experiments include the nFST, WARM and nFST_WARM simulations (Table 1). nFST is identical to REF but without fast ice representation i.e., no tensile strength parameterisation and no grounded icebergs representation. WARM and nFST_WARM have the same setup as REF and nFST, respectively, but cover the 2081-2100 period. In these simulations,

the model is forced by climate anomalies derived from a climate change projection carried out with the global climate model EC-Earth3 under the SSP4-4.5 scenario (Döscher et al., 2021), within the 6th phase of the Coupled Model Intercomparison Project (Eyring et al., 2016). Note that, in WARM, the grounded icebergs location are the same as in REF and that, as for REF and nFST, WARM and nFST_WARM are run from a 20-yr spin-up. Two more sensitivity experiments have been conducted to disentangle the effects of both the atmospheric and oceanic forcings on the ASC acceleration. WARM_noAtm is similar to WARM, except that this simulation has no EC-Earth3 anomaly applied to the atmospheric forcing (the atmospheric forcing is thus identical to the one in REF). WARM_noOce is equivalent to WARM but without EC-Earth3 anomalies applied to the ocean velocities at the lateral boundaries of the model domain.

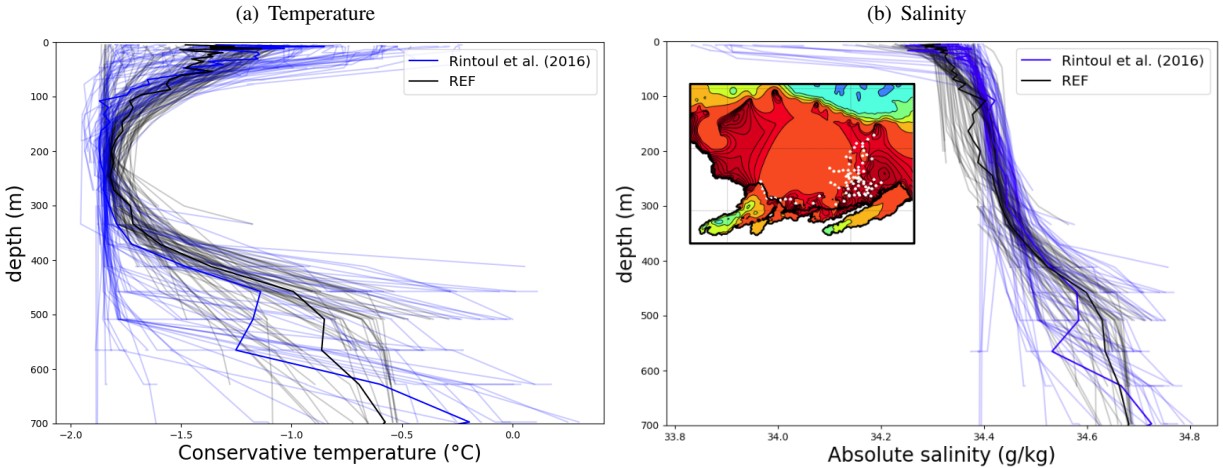

**Figure 2.** Vertical profiles of temperature (a) and salinity (b) after the bias correction on the continental shelf in front of the Totten ice shelf. Blue: CTD from Rintoul et al. (2016) (a1402). Black: as simulated in REF. Simulated profiles are taken at the same time and location as the CTD measurements. The observations have been collected in two locations, close to the TIS front and near the Dalton coastal polynya. The locations are denoted by white dots in the panel displayed in subfigure b.

Annual cycles of the EC-Earth3 climate anomalies are computed as the differences between 2081-2100 and 1995-2014, and are added to all the fields of the atmospheric and oceanic forcings used for the 1995-2014 period in REF and nFST (for the atmosphere: wind velocity, temperature, specific humidity, surface downward radiation and precipitation; for the ocean: current velocity, temperature, salinity, sea surface height, sea ice concentration, sea ice thickness and snow thickness). Figure 3 shows the annual mean ocean temperature, salinity and zonal ocean velocity anomalies at the eastern boundary condition, and the mean near-surface (2 m) air temperature and atmospheric zonal wind (10 m) velocity anomalies. We show the ocean anomalies at the eastern lateral boundary condition as they are very similar to those at the western lateral boundary condition, and also because the ocean eastern boundary condition is one of the drivers of the ocean dynamic over the continental shelf in regional modelling (Nakayama et al., 2021). The ocean temperature anomaly is positive everywhere, with values from 0 to $0.5°$ C over the continental shelf and in the deep ocean, and from 1 to $1.5°$ C in the upper ocean outside of the shelf. The seawater salinity anomaly is mostly negative (down to -0.4 g/kg), with the lower values above the continental shelf. Oceanic zonal velocity

anomalies at the eastern boundary are westward over the shelf and eastward off the shelf. The EC-Earth3 anomaly applied at the zonal wind component is mostly eastward over the ocean, increasingly towards the north. Westward wind anomalies also occur, but only over a small part of the shelf and over the continent. The surface air temperature anomaly is positive everywhere (Fig 3e), with values larger than $1°$ C and up to $1.8°$ C near the coast.

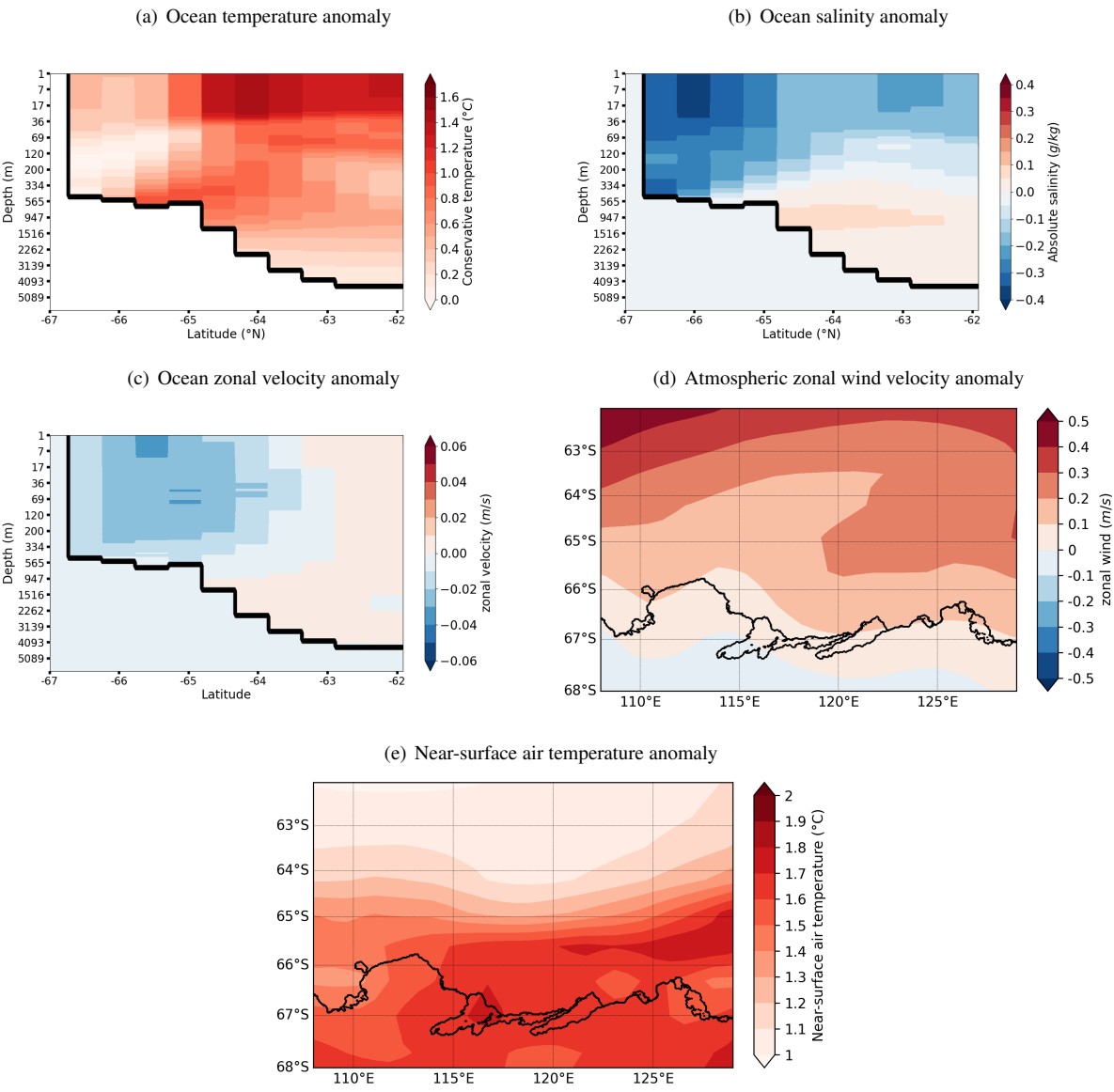

**Figure 3.** Annual mean EC-Earth3 anomalies applied at the eastern boundary of the model domain for the conservative temperature (a), absolute salinity (b) and the zonal component of the ocean velocity (c). Annual mean EC–Earth3 anomaly of the wind velocity (10 m) zonal component (d) and the near-surface (2 m) air temperature (e). The anomaly are computed between the 2081-2100 and the 1995-2014 periods.

| | Landfast ice | Forcing and lateral boundary conditions |
|---|---|---|
| REF | yes | Recent past (ERA5, PARASO, 1995-2014) |
| WARM | yes | REF + anomalies derived from EC–Earth3 climate change projection |
| nFST | no | Recent past (ERA5, PARASO, 1995-2014) |
| nFST_WARM | no | REF + anomalies derived from EC–Earth3 climate change projection |
| WARM_noAtm | yes | WARM - atmospheric anomalies derived from EC–Earth3 climate change projection |
| WARM_noOce | yes | WARM - ocean velocity anomalies derived from EC–Earth3 climate change projection |

**Table 1.** Names and descriptions of the simulations used in this study.

## 3   Results

In this section, we examine the main differences between the results from the REF and WARM simulations. Figures 4a and 4b display the geographical distribution of the fast ice frequency, defined as the percentage of days in a year with a 2-week mean sea ice velocity lower than 0.005 m/s. There is a large retreat of fast ice in WARM compared to REF in front of both the TIS and MUIS. In front of the TIS, the multiyear fast ice cover (frequency above 0.9) in REF is replaced by first year fast ice in WARM. On the other hand, the first year fast ice (frequency between 0.4-0.8) in REF is no longer present in WARM. The same frequency decrease occurs in front of the MUIS, where most of the multiyear fast ice in REF becomes first year fast ice in WARM, with a 50% frequency reduction, and the first year fast ice in REF has vanished in WARM. This loss of the multiyear fast ice in WARM is mainly due to the atmospheric forcing, as hinted by the fast ice simulated in WARM_noAtm presented in Figure 4c, which is much closer in both frequency and area to the fast ice simulated in REF than in WARM.

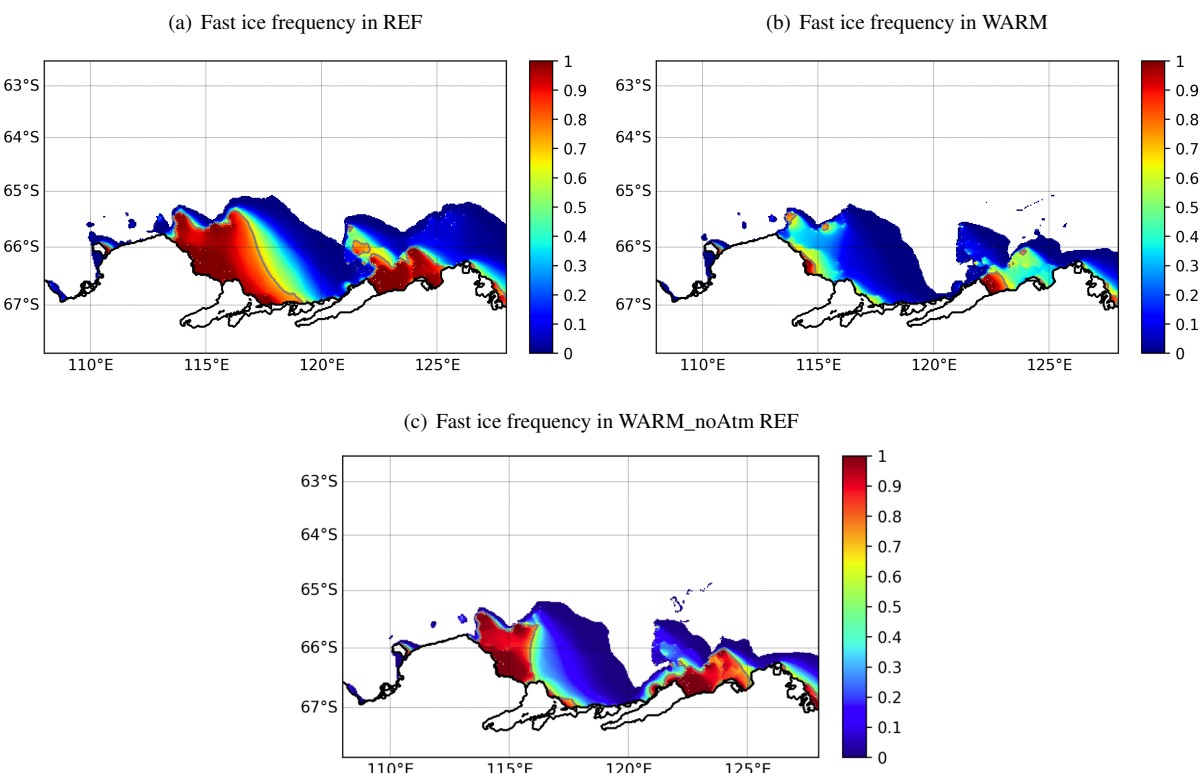

**Figure 4.** Fast ice frequency for the REF (a), WARM (b) and WARM_noAtm (c) simulations, all averaged over the 20 years of simulation. The 0.75 fast ice frequency is shown by the gray line in the fast ice frequency.

As shown by Figure 5c to 5f, the changes in sea ice concentration over the continental shelf between REF and WARM mostly occur during summer months, where the loss of multiyear fast ice reduces the sea ice extent and concentration. In winter, changes are limited to the region off the continental shelf, with a general southward retreat of the ice edge in WARM.

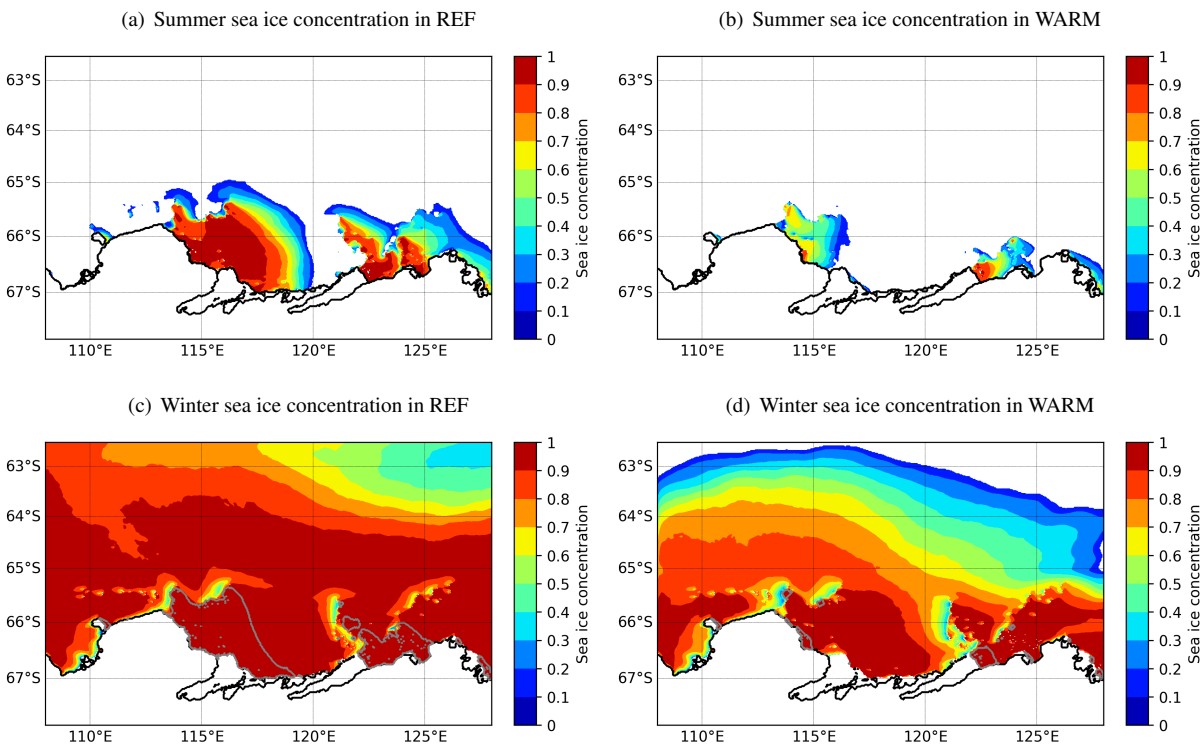

**Figure 5.** Sea ice concentration in summer (JFM) and winter (JASO) for the REF (left) and WARM (right) simulations, both averaged over the 20 years of simulation. The 0.75 fast ice frequency is shown by the gray line in the winter sea ice concentration map.

175      The differences in mean sea ice production between both simulations (Fig. 6) exhibits the important changes in sea ice formation related to the fast ice changes presented above. The partial disappearance of multiyear fast ice in WARM induces stronger interactions (heat fluxes) between the cold atmospheric air and the ocean surface, which increases the sea ice production near the coast. This increase of sea ice production along the coast in WARM is counterbalanced by the decrease in sea ice production off shore, on the western side of the large fast ice packs that are present in REF but not in WARM.

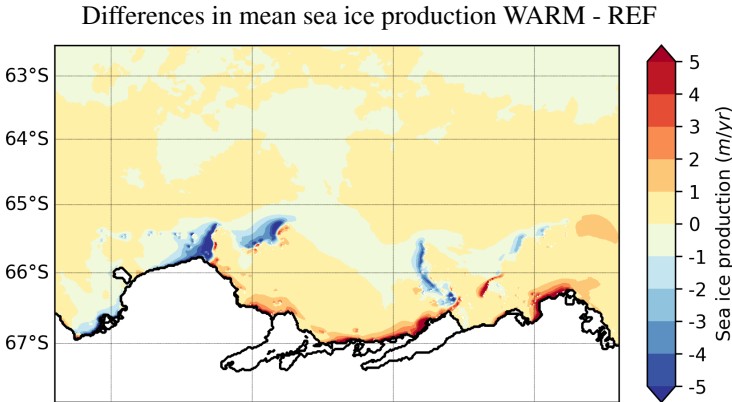

**Figure 6.** Differences in mean sea ice production between WARM and REF, averaged over the 20 years of simulation. Positive values mean that WARM has more sea ice production than REF.

Figures 7a and 7d reveal that the ocean circulation experiences major changes between REF and WARM. The ASC, which is barely present in REF, is strongly enhanced in WARM, especially in front of Law Dome and in front of the MUIS (the mean ocean velocity at the ASF is less than 0.1 m/s in REF and is close to 0.15 m/s in WARM). Furthermore, the Totten oceanic gyre in front of the TIS (clockwise oceanic circulation over the shelf) is intensified in WARM, especially its western and southern components. This acceleration mainly results from the retreat of fast ice, which acts as a dynamically isolating cover that inhibits the transmission of wind stress to the ocean. Indeed, the mean ocean barotropic velocities in nFST and WARM (Fig. 7b and 7d) present the same pattern and intensity near the coast, which suggests that the ocean current differences near the coast between WARM and REF are only due to the loss of fast ice in WARM. Moreover, as the mean ocean barotropic velocities in front of Totten in both WARM_noOce and WARM are similar, this confirms that the change in ASC intensity is not the source of this coastal current acceleration. The integrated ocean volume transport at the southern edge of the gyre, near the front of the TIS cavity is increased by $226\%$ in WARM compared to REF (from 0.55 to 1.8 Sv). This accelerated gyre speeds up the ocean masses entering the TIS cavity, which contributes to the increased basal melting. Figure 7e shows the annual mean, depth-integrated zonal oceanic volume transport for the REF, WARM, WARM_noAtm and WARM_noOce simulations. For each simulation, this mean transport is westwards everywhere (positive value) from the coast until 63°S, with a maximum value near 65°S where the ASC is located (at the shelf break). The eastward transport north of 63°S is associated with the Antarctic Circumpolar Current (ACC). REF and WARM exhibit the same transport pattern, but with a $116\%$ increase of the ASC in WARM compared to REF.

As suggested by the similar patterns of westward ocean transport in both WARM and WARM_noAtm (see Fig. 7e), the ASC intensification in WARM is not wind-driven. Indeed, as the pressure gradient across the ASF is enhanced by easterly winds that drive the sea surface height gradient through Ekman drift (Mathiot et al., 2011), an ASC intensification would required stronger easterly winds. Nevertheless, the EC-Earth3 wind velocity anomalies applied to the model in WARM are mostly positive (Fig. 3e), which weakens the easterly winds. The ASC intensity difference between WARM and WARM_noOce (between 65.6 and

64.7°S) in Figure 7e indicates that the ocean velocity anomalies derived from EC-Earth3 and applied to the oceanic forcing in WARM are responsible for 83% of the ASC increased intensity between REF and WARM. The remaining 17% of ASC increased intensity between REF and WARM, which corresponds to the difference in ASC magnitude between REF and WARM_noOce, could have a density-driven origin, as the lateral density gradient across the ASF contributes to establishing the geostrophically balanced, vertically sheared along-slope flows of the ASC (Lockwood et al., 2021). This is coherent with the large density lowering over the continental shelf in WARM compared to REF, which leads to a stronger density gradient across the ASF (Fig. 7f). Since the seawater density is mostly a function of salinity in the Southern Ocean (Pellichero et al., 2018), the ASC modification should then be linked to the changes in sea ice production and melt occurring in WARM. These changes, in addition to the EC-Earth3 salinity anomalies prescribed at the eastern boundary of the domain (Fig. 3b), reduce the ocean salinity over the shelf.

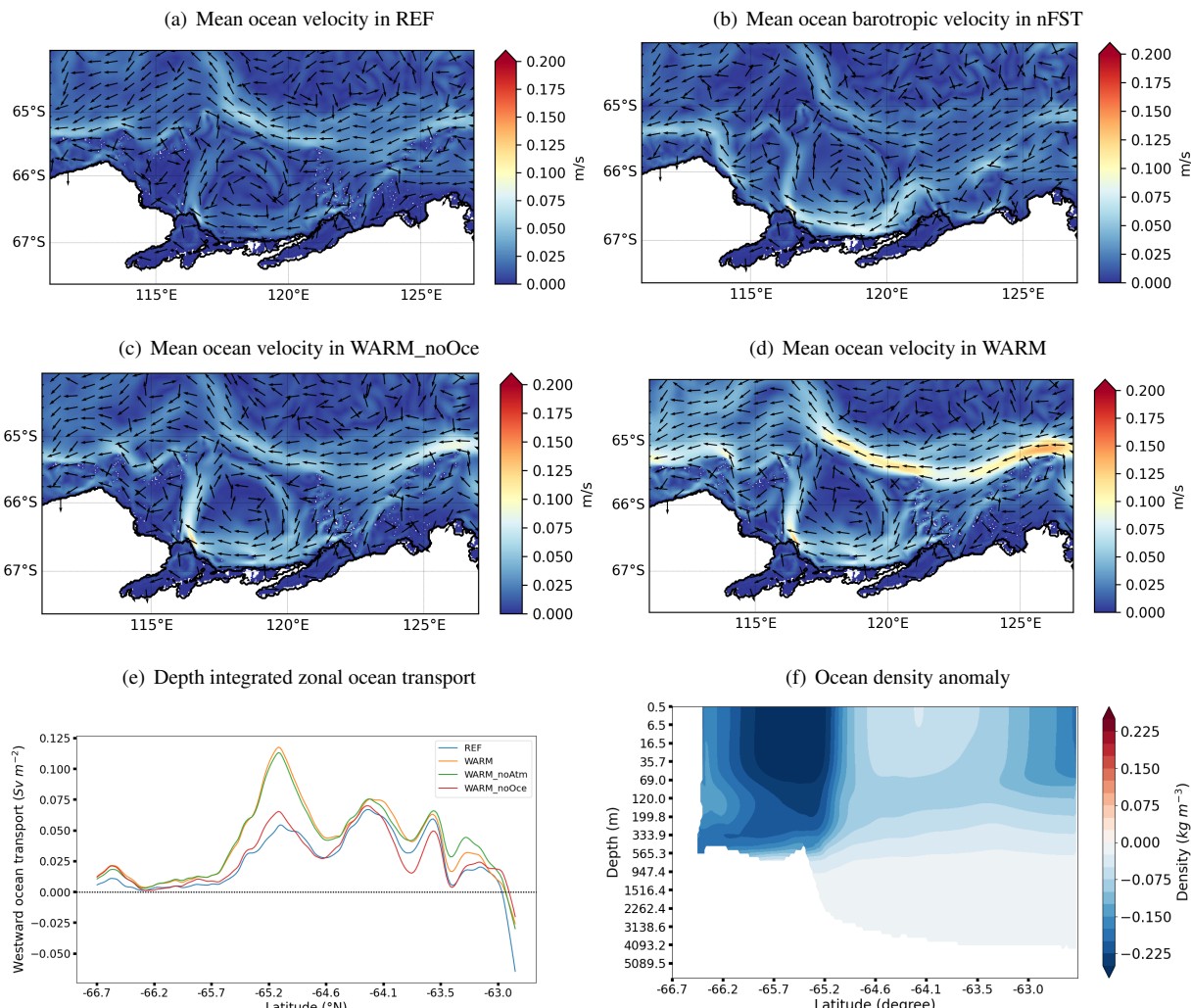

**Figure 7.** Annual mean, depth-averaged ocean velocity for the REF (a), nFST (b), WARM_noOce (c) and WARM (d) simulations, both averaged over the 20 years of simulation. (e) Annual mean, depth-integrated zonal ocean volume transport for the REF, WARM, WARM_noAtm and WARM_noOce simulations. (f) Meridional section of the ocean density change between WARM and REF.

As hinted by Nakayama et al. (2021), at equivalent oceanic and atmospheric warmings, the ASC modulates the heat intrusion towards the continental shelf and the TIS and MUIS cavities. The basal melt rate for both cavities in WARM and WARM_noOce (Fig. 8) shows higher values with low ASC intensity (WARM_noOce) and lower values with high ASC intensity (WARM). This implies that, whereas the ocean and surface air temperature increase induces the intrusion of warmer water into the ice shelf cavities and higher basal melt rate, the accelerated ASC limits this basal melt rate increase. However, this ASC effect is hidden in Figure 9 by the ocean warming due to the atmospheric and oceanic forcings.

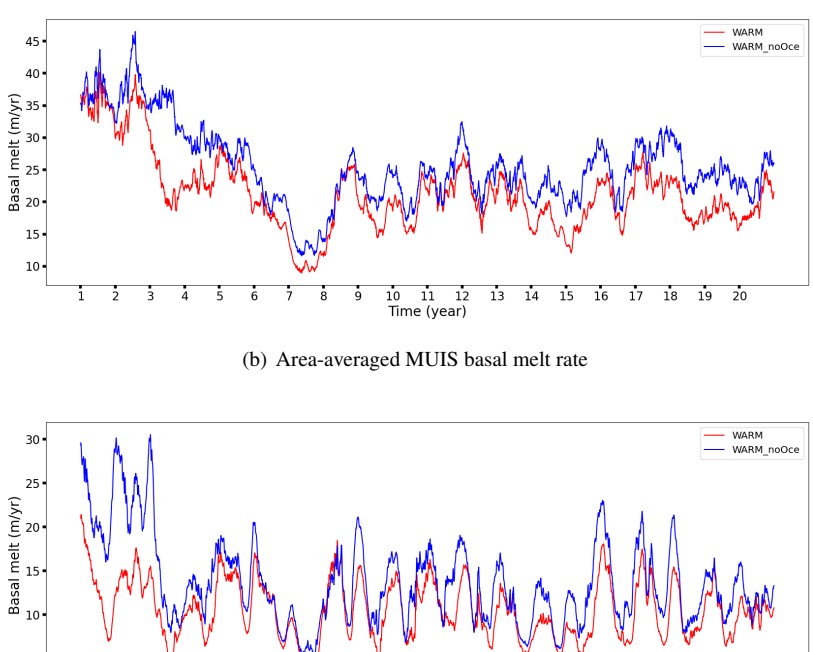

(a) Area-averaged TIS basal melt rate

(b) Area-averaged MUIS basal melt rate

**Figure 8.** Time series of the area-averaged TIS (a) and MUIS (b) basal melt rates from WARM (red) and WARM_noOce (blue) for the 20 years of simulations.

Figure 9 depicts the annual mean ocean temperature differences between WARM and REF (WARM - REF) over the continental shelf at 200, 300, 400 and 500 m depth. Despite an intensified ASC, which tends to isolate the continental shelf from the open ocean by reducing the across-shelf exchanges, the ocean temperature over the continental shelf in WARM features an overall increase. Figure 9a shows warmer water mostly everywhere at 200 m, with a slight warming (from 0.1 to 0.4° C) over the shelf and a larger warming (from 0.4 to 1° C) in the open ocean. Cooler waters are found on the eastern flank of the MUIS cavity (from 0 to $-0.2°$ C). The same pattern of temperature difference is noticed at 300 and 400 m (Fig. 9b and 9c), with a slight cooling next to MUIS and a strong warming in front of TIS, inside the Totten oceanic gyre, where the temperature increase reaches more than $+1°$ C. Deeper, at 500 m, the temperature difference in front of the MUIS becomes positive (up to $+0.2°$ C), and the cooling in front of the MUIS is now restricted to the region east of 126°E (Fig. 9d). The difference of ocean warming between the front of the TIS and the front of MUIS is mostly due to the differences in bathymetry in the two areas. Indeed, both ice shelves present the same warmer ocean masses at the shelf break but the deeper bathymetry in front of the TIS (up to 600 m) allows more warming to reach the TIS cavity.

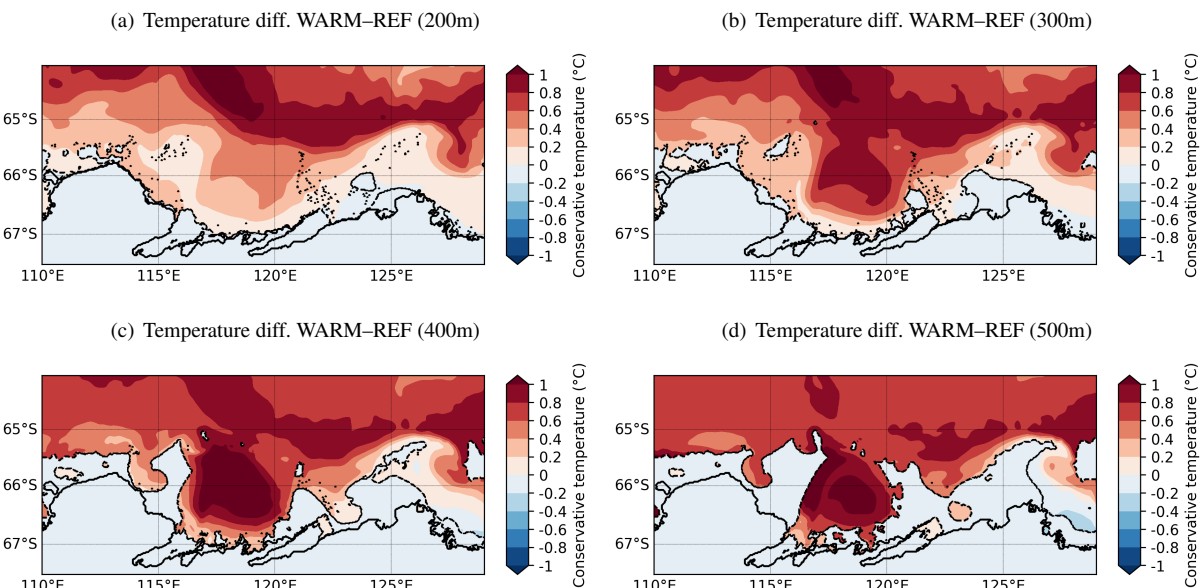

**Figure 9.** Annual mean ocean temperature differences between the WARM and REF simulations over the continental shelf at 200, 300, 400 and 500 m depths, all averaged over the 20 years of simulation. The dashed line depicts the contours of the 200, 300, 400 and 500 m depth topography for fig a, b, c and d, respectively.

Finally, Figures 10a and 10b display the area-averaged ice shelf basal melt rate for both the TIS and MUIS from REF and WARM, respectively. The TIS experiences a larger (+91%) and more variable (+130% in standard deviation) basal melt rate in WARM compared to REF. By contrast, the MUIS basal melt rate exhibits a lower basal melt rate increase (+36% increase in WARM) and a lower basal melt rate variability increase (+33% in standard deviation). The lower basal melt increase in MUIS can be attributed to the lower ocean warming in front of the MUIS cavity, with less than $+0.2°$ C in front of MUIS compared

to more than $+1°$ C in front of the TIS (see Figure 9). The drop of basal melt rate in the sixth and seventh years is due to the ocean boundary conditions. Figure 10c shows the differences in spatial distribution of the mean basal melt rate inside the TIS and MUIS cavities between the REF and WARM simulations. The melt rates increase spans from few meters per year to more than 45 meters of ice per year. The highest basal melt rate increase between REF and WARM are located on the western side of each cavities, near the grounding line, where the ocean circulation within the cavities is the fastest (up to +45m/yr in Totten

and up to +20 m/yr in MUIS).

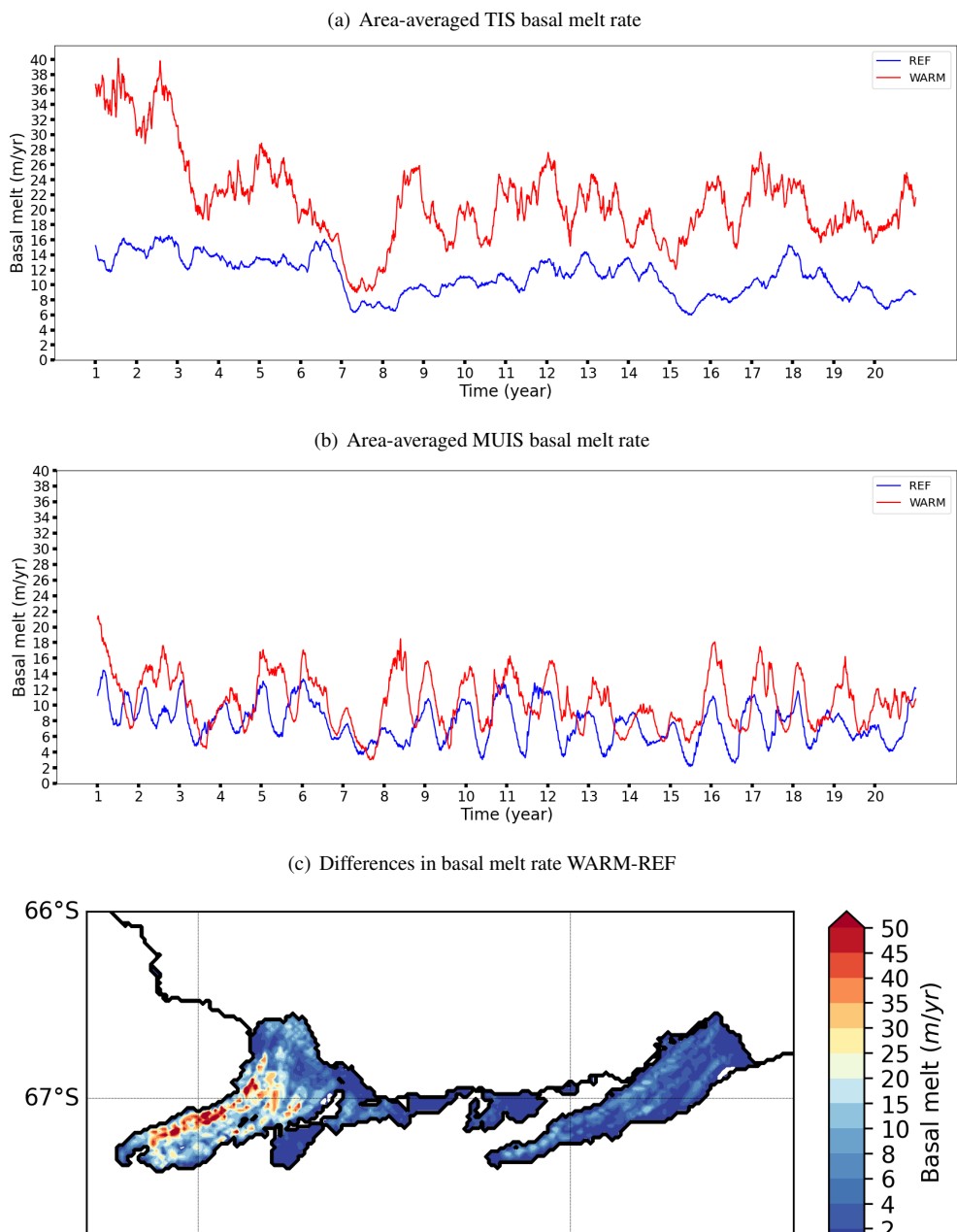

**Figure 10.** Time series of the area-averaged TIS (a) and MUIS (b) basal melt rates from the REF (blue) and WARM (red) simulations. Spatial distribution of the differences in ice shelf basal melt rate between the REF and WARM simulations. The time periods are 1995-2014 for REF and 2081-2100 for WARM. The mean TIS basal melt rate is $11.13 \pm 2.54$ m/yr in REF and $21.29 \pm 5.88$ m/yr in WARM, while the MUIS basal melt rate is $7.73 \pm 2.51$ m/yr in REF and $10.51 \pm 3.35$ m/yr in WARM.

The increased temporal variability of both TIS and MUIS basal melt rates in WARM is not related to the loss of fast ice (see Tab. 2), but could be explained by the larger Mixed Layer Depth (MLD) variability in front of the cavities in WARM (see Fig. 11). Due to the MLD effect on the ocean stratification and on the intrusion of warm water into the cavities (Van Achter et al., 2022), this stronger MLD variability, which is related to the larger amplitude of the surface air temperature seasonal cycle, increases the variability of warm water intrusion into the ice shelf cavities.

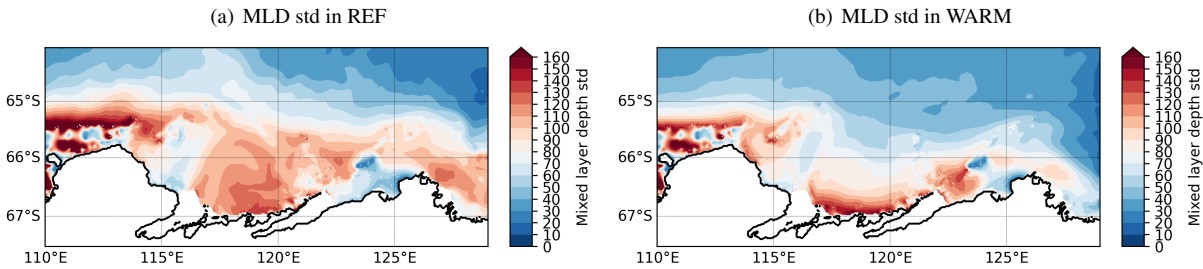

**Figure 11.** Standard deviation of the mixed layer depth for both the REF (a) and WARM (b) simulations.

## 4  Ice shelves melt rate sensitivity to fast ice in a warming climate

In this section, we analyse how the presence of fast ice, implemented through the combination of both a sea ice tensile strength parameterisation and a representation of grounded icebergs, impacts the changes in ice shelf basal melt rate between the recent past and the end of the 21st century. The area-averaged TIS and MUIS basal melt rates for both nFST and nFST_WARM are shown in Figure 12. The TIS has a basal melt rate of $8.74 \pm 2.76$ m/yr and $20.68 \pm 5.87$ m/yr in nFST and nFST_WARM, respectively, whereas the MUIS has a mean basal melt rate of $6.28 \pm 2.25$ m/yr and $11.01 \pm 4.67$ m/yr in nFST and nFST_WARM, respectively.

| Ice shelves | Fast ice | Recent past (1995-2014) | End of the 21st century (2081-2100) |
|---|---|---|---|
| Totten | yes | $11.13 \pm 2.54$ m/yr | $21.29 \pm 5.88$ m/yr (+91%) |
| | no | $8.74 \pm 2.76$ m/yr | $20.68 \pm 5.87$ m/yr (+136%) |
| Moscow | yes | $7.73 \pm 2.51$ m/yr | $10.51 \pm 3.35$ m/yr (+36%) |
| University | no | $6.28 \pm 2.25$ m/yr | $11.01 \pm 4.67$ m/yr (+75%) |

**Table 2.** Mean ice shelf basal melt rates for both the recent past and the end of the 21st century and for all simulations.

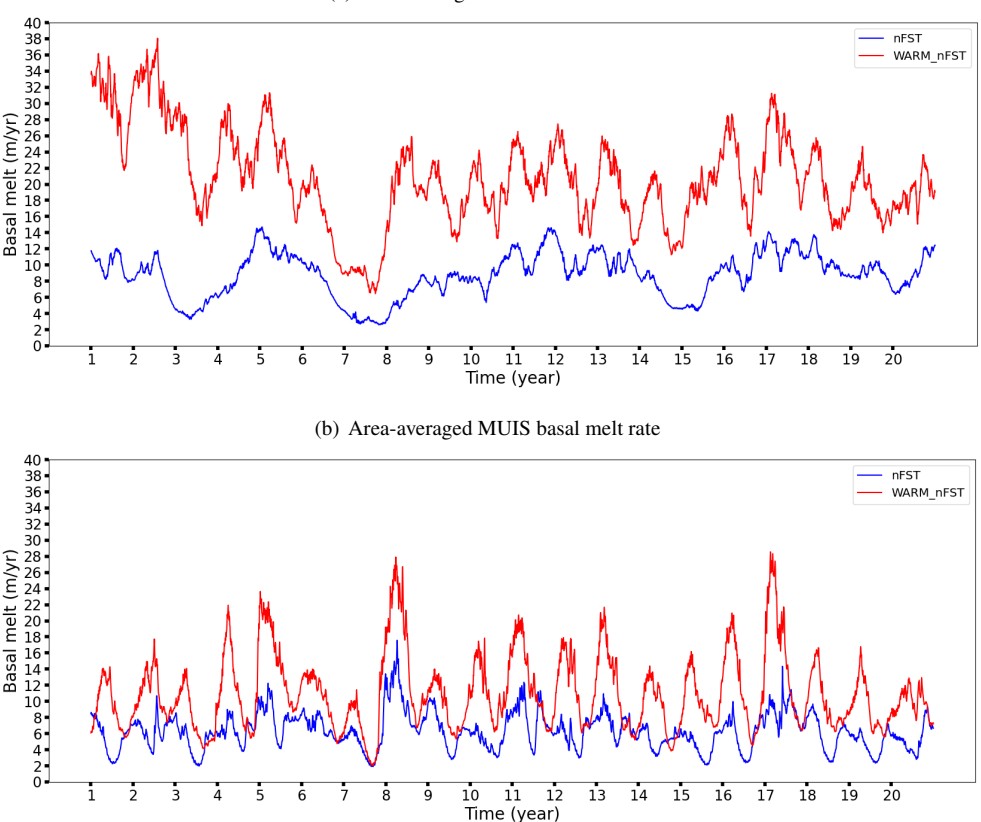

(a) Area-averaged TIS basal melt rate

(b) Area-averaged MUIS basal melt rate

**Figure 12.** Time series of the area-averaged TIS (a) and MUIS (b) basal melt rates from nFST (blue) and nFST_WARM (red). The timescale are 1995-2014 and 2018-2100 for the recent past simulations (blue) and the future climate conditions (red), respectively. The time period are 1995-2014 for nFST and 2081-2100 for nFST_WARM. TIS melt rate are $8.74 \pm 2.76$ m/yr in nFST and $20.68 \pm 5.87$ m/yr in nFST_WARM. MUIS melt rate are $6.28 \pm 2.25$ m/yr in nFST and $11.01 \pm 4.67$ m/yr in nFST_WARM.

The mean melt rates at the base of the TIS and MUIS for all simulations are given in Table 2. Without fast ice representation, the increase in basal melt rate for both ice shelves between the two time periods is much larger. This is explained by both

the strong impact of fast ice on the ice shelf basal melt rate for the recent past simulation (difference of more than 1.45 m/yr between REF and nFST) and by its small impact on the ice shelf basal melt rate by the end of the 21st century (difference of less than 0.6 m/yr between WARM and WARM_nFST). The strong fast ice impact on the basal melt rate in the recent past simulations is related to the displacement of the sea ice production zones (see Fig. 13), by the fast ice, from coastal to offshore areas. This change in sea ice production induces less sea ice production and more sea ice melt near the coast, which increases

the ocean stratification in front of the cavities, favors warm water intrusions and increases the ice shelf basal melt rate in REF compared to nFST (Van Achter et al., 2022). However, as the fast ice shrinks under warmer oceanic and atmospheric conditions of the end of the 21st century (Fig. 4a and 4b), this fast ice impact on the ice shelf basal melt rate is strongly reduced. So, with lower ice shelf melt rates in nFST than in REF but with no significant melt rate changes between WARM and nFST_WARM,

the simulations without a fast ice representation are showing a stronger ice shelf melt rate growth between the two periods. In other words, the effect of the reduced extent of fast ice on the ice shelf basal melt rate offsets part of the melt rate increase due to warmer atmospheric and oceanic conditions.

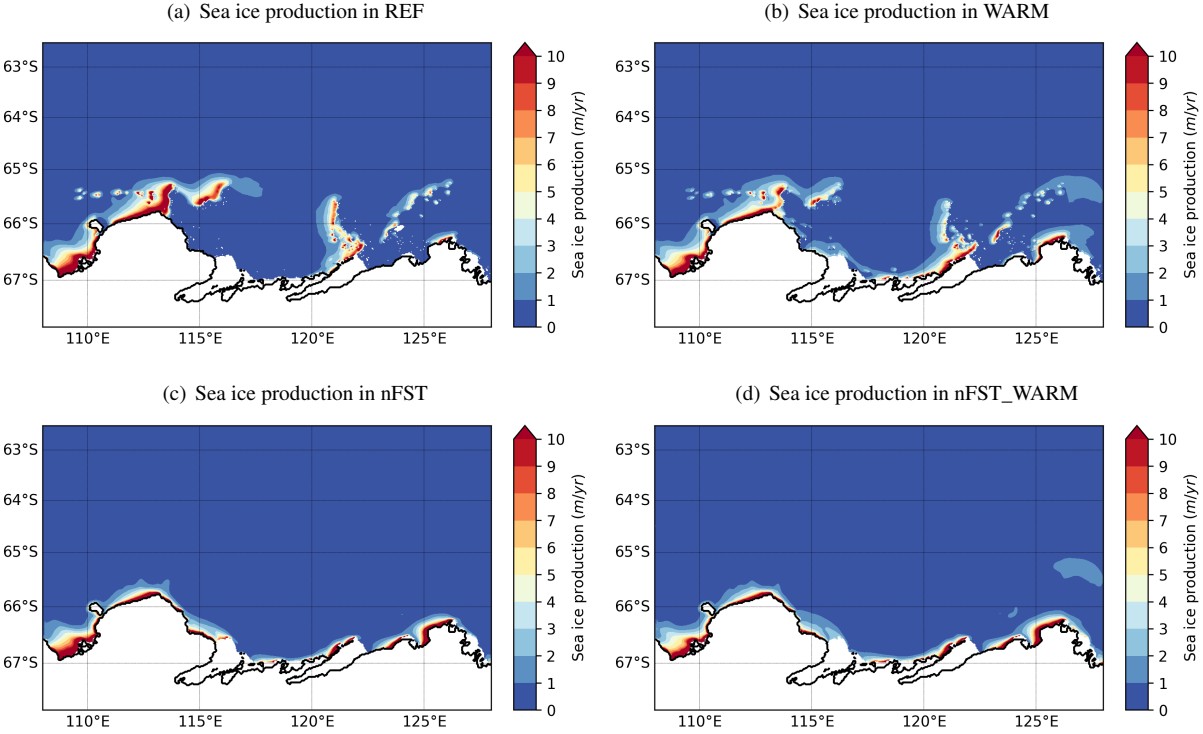

**Figure 13.** Mean sea ice production for the REF (a), WARM (b), nFST (c) and nFST_WARM (d) simulations, all averaged over the 20 years of simulation.

Finally, the higher MUIS basal melt rate in nFST_WARM compared to WARM is attributed to the changes affecting the sea ice in WARM and nFST_WARM. In nFST_WARM, the absence of fast ice allows strong sea ice formation along the coast, with a deep MLD in front of the MUIS cavity (Fig. 14c). In contrast, in WARM, the presence of fast ice allows for sea ice formation at the off-shore polynya created on the west side of fast ice patches in front of the MUIS cavity, but it also allows strong sea ice production along the coast since the fast ice there is largely reduced in area and frequency. This combination of sea formation both off-shore and along the coast contributes to a broader area of deep MLD in front of the MUIS cavity in WARM (Fig. 14d), which decreases the amount of warm water able to cross the continental shelf and to reach the MUIS cavity in WARM compared to nFST_WARM (Fig. 14a and 14b). As a consequence, the MUIS basal melt rate in WARM is lower than in nFST_WARM.

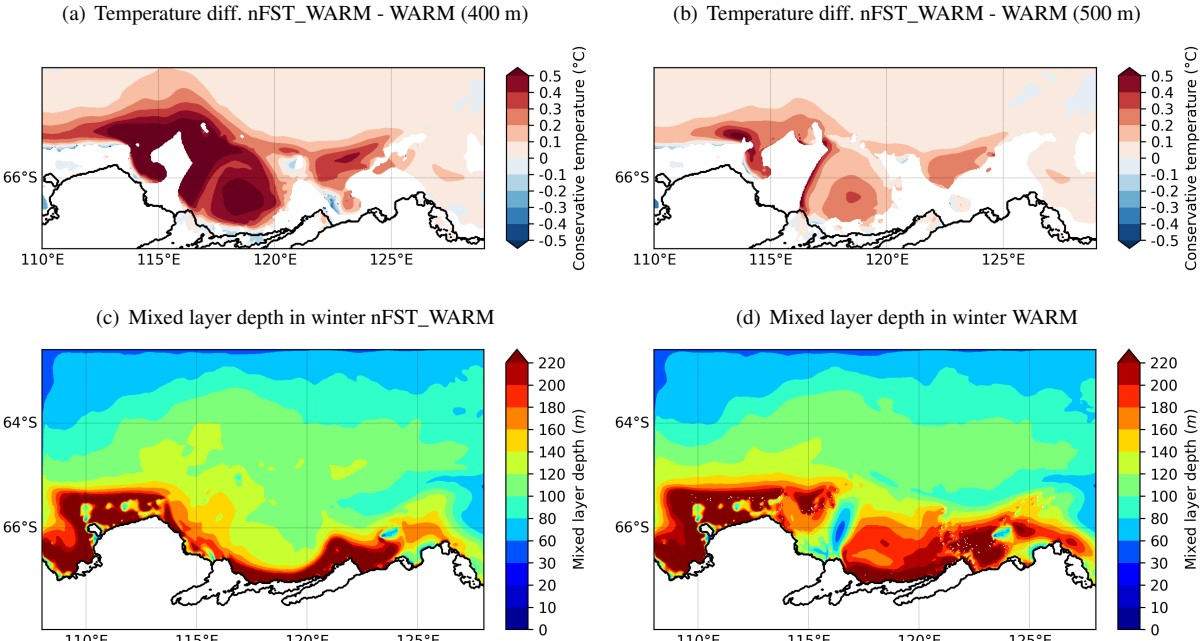

**Figure 14.** Annual mean ocean temperature differences between the nFST_WARM and WARM simulations over the continental shelf at 400 (a) and 500 m depths (b). Annual mean MLD for nFST_WARM (c) and WARM (d) for the winter months (JASO). Both the temperature anomalies and the MLD are averaged over the 20 years simulation.

## 5 Discussion and conclusions

The first goal of this study was to investigate the ocean–ice shelf interactions under warmer climate conditions in the Totten Glacier region. To do so, we applied climate anomalies, obtained from a SSP4-4.5 climate change projection conducted with EC-Earth3, at the oceanic boundary conditions and atmospheric forcing of a NEMO-LIM high-resolution, regional config-
uration, which includes an explicit treatment of ocean–ice shelf interactions and a fast ice representation. Our experiments revealed major changes in ice shelf basal melt rate, sea ice production and ocean circulation between recent past (1995-2014) and the end of the 21st century (2081-2100). The sea ice extent is reduced in both summer and winter, with a general southward retreat of the ice edge. The fast ice forms less frequently and its coverage is strongly reduced. Both TIS and MUIS undergo a drastic basal melt increase, with a $91\%$ and $36\%$ increase, respectively. Such changes in the ice shelf basal melt rate can be
attributed to warmer mCDW, with more than $+1°C$ of ocean warming in front of the TIS cavity and up to $+0.2°C$ in front of the MUIS cavity. The warmer ocean conditions have a lesser effect on the MUIS basal melt rate, mainly because of the shallower bathymetry in front of its cavity, but also because of the accelerated gyre in front of the TIS cavity, whose acceleration is due to the disappearance of fast ice. This accelerated gyre speeds up the ocean masses entering the TIS cavity and contributes to the basal melt rate increase. In the ocean, the ASC is intensified, with an oceanic zonal volume transport that is increased by
$116\%$ in WARM compared to REF. This strengthening of the ASC is attributed to both the EC-Earth3 ocean velocity anomalies

applied to the ocean forcing (83%) and to the changes in density gradient (mostly salinity) across the shelf (17%), triggered by both the sea ice production modification and the salinity changes in the ocean lateral boundary conditions. The accelerated ASC reduces the cross-slope water exchanges and tends to decrease the melt rate in both ice shelf cavities, partly compensating the effect of the oceanic warming.

The second goal of this study was to determine how fast ice influences the increase in ice shelf basal melt rate between the recent past and the end of the 21st century. The representation of fast ice, through the combination of both a sea ice tensile strength parameterisation and the representation of grounded icebergs, has been shown to offset the basal melt rate increase simulated between the recent past and the end of the 21st century. Indeed, for the TIS, the average recent past melt rates exhibit a strong sensitivity to fast ice, with higher melt rate with the fast ice representation and show a lower role of the fast ice by the

end of the 21st century. For MUIS, the situation is similar, except that the end of the 21st century basal melt rate is slightly lower with the fast ice representation due to spatial changes in the MLD. The fast ice impact on the melt rate drops as the fast ice extent is reduced due to the warmer oceanic and atmospheric conditions by the end of the 21st century. So, with higher melt rate values for the recent past and with closer melt rate values by the end of the 21st century, the simulations with fast ice have a lower melt rate trend between the two periods than the simulations without a fast ice representation. This enlightens the

importance of fast ice, either for studying melt rate by the end of the 21st century alone, or for studying the evolution of basal melt rate across the 21st century.

Few other studies investigate the ice shelf melt rate increase between present days and the end of the 21st century in the Totten Glacier area. Moreover, the amount of melt rate increase is strongly linked to the model, initial conditions and climate change scenario used to force the model. Furthermore, as recent studies are suggesting both strengthening and weakening of

the ASC in the future (Moorman et al., 2020; Pelle et al., 2021), we should aim for better understanding of the ASC changes in East Antarctica.

One of the main limitations of our study lies in the lack of knowledge about the grounded iceberg distribution by the end of the 21st century. In the absence of a day-to-day high-resolution iceberg map, we used a 2-month iceberg dataset (September–October months of 1997) to prescribe the grounded iceberg location for both the REF (1995-2014) and WARM

(2081-2100) simulations. However, a change in the iceberg distribution between REF and WARM might influence the results presented here. Indeed, a modification of the iceberg density in front of the TIS and MUIS cavities could either increase or decrease the fast ice distribution over the continental shelf, and consequently influence how the fast ice change damps the ice shelf basal melt rate under warming conditions. Another limitation in our experimental design, is the use of only one climate change projection. As the ice shelf basal melt rates at the end of the 21st century show no significant sensitivity to the fast ice,

the use of a stronger climate change scenario was not relevant for this research. However, the effect of a stronger scenario on the ASC would be interesting. Still about the experimental design, the REF and WARM simulations have the same interannual variability (because WARM is REF with EC–Earth3 anomalies). A WARM simulation with its own interannual variability might change how the TIS and MUIS basal melt rates are enhanced in WARM. Moreover, since these results are strongly linked to local processes, it would be interesting to look at the same mechanisms but in other regions of East Antarctica. Finally, as

the easterly wind component is projected to weaken over the next century and will significantly impact the Southern Ocean circulation (Neme et al., 2022), the ASC change analysis should be extended to a wider scale and to other regions.

    Overall, the ASC acceleration and its effect on the basal melt rate highlight the benefits of high-resolution and accurate continental shelf bathymetric datasets in order to represent lateral density gradients associated with the ASF, and thus to simulate realistically the ASC. This is a major challenge for global climate models, whose relatively coarse resolution prevents

such phenomena from being accurately represented (Lockwood et al., 2021). Furthermore, our results underline the worth of a prognostic fast ice representation to simulate ice shelf melt rate evolution in Antarctica. In contrast to the prescribed fast ice, the prognostic approach enables the fast ice extent to evolve in time (Nihashi and Ohshima, 2015; Van Achter et al., 2022). The prognostic representation of fast ice, with time-evolving grounded iceberg locations should be one of the key focus in high-resolution ocean-sea ice modelling in East Antarctica for the years to come.

*Author contributions.* GVA designed the science plan with TF and HG, ran the simulations, produced the figures, analysed the results and wrote the manuscript based on insights from all co-authors. EMC provided the EC–Earth3 dataset.

*Competing interests.* The authors declare that they have no known competing financial interests or personal relationships that could have appeared to influence the work reported in this paper.

*Acknowledgements.* We thank the editor and the two referees for their comments and suggestions. This work was supported by the PARAMOUR
project, "Decadal predictability and variability of polar climate: the role of atmosphere-ocean-cryosphere multiscale interactions", supported by the Fonds de la Recherche Scientifique – FNRS and the FWO under the Excellence of Science (EOS) program (grant no. O0100718F, EOS ID no. 30454083). HG is research director with the F.R.S-FNRS (Belgium). Computational resources have been provided by the supercomputing facilities of the Université catholique de Louvain (CISM/UCL) and the Consortium des Equipements de Calcul Intensif en Fédération Wallonie Bruxelles (CECI) funded by the Fond de la Recherche Scientifique de Belgique, Belgium (F.R.S.-FNRS) under conven-
tion 2.5020.11. The present research benefited from computational resources made available on the Tier-1 supercomputer of the Fédération Wallonie-Bruxelles, infrastructure funded by the Walloon Region under the grant agreement n1117545.

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
