# Peer review of "Influence of fast ice on future ice shelf melting in the Totten Glacier area, East Antarctica"

_EGUsphere, 2022_

## Author Response (AR1)

**Response to the Referee #1**

Dear Referee,

Thank you for the time that you spent on our manuscript. Below you will find a summary of the changes that we made throughout the manuscript to address all your suggestions.

Yours sincerely

On behalf of all the co-authors,

Guillian Van Achter

**General comments**

This study used a high-resolution (2km) regional ocean-sea ice-ice shelf model to investigate the responses of landfast ice, sea ice, ice-shelf basal melt, and ocean around the Totten Ice Shelf (TIS) to a future warming climate scenario (SSP4-4.5). The novelty of this study is applying the prognostic fast ice component that the authors developed as a part of a sea-ice model component in their previous study. Although I have several concerns and suggestions, I think that this paper will be suitable for publishing in The Cryosphere after substantial revision.

**Specific comments**

1. [Major] L9-11 "The representation of fast ice ..."and discussions with Table 2. This study concludes that the response of ice-shelf basal melting at the Totten Glacier becomes prominent in the experiments with landfast ice, compared to those without landfast ice. I think that the conclusion is slightly misleading. The areal extent of fast ice becomes small under the future warming condition, and there are no significant differences in the Totten Glacier melting between the numerical experiments with and without fast ice. A large difference in the TIS basal melting is only found in the present-day (1995-2014) condition, creating the tendency in the experiments with and without fast ice.

We changed the abstract, the results section and the conclusion to avoid further confusion. As you have summarised, there are no significant effect of the fast ice on the basal melt rate in the end of the 21st century. The effect of fast ice on the melt rate is important in the last decades simulation but then becomes negligible with the decrease of sea ice cover during the 21st century (see page 1 lines 9-16, pages 15-16 lines 244-257, page 17-18 lines 286-297).

2. [Major] The literature, Pelle et al. (2021), used a high emission scenario, but this study used the moderate one, SSP4-4.5, without any explanation/motivation. If possible, I strongly recommend performing additional experiments under high emission scenarios to compare the previous study and obtain more solid results under warming climates.

We acknowledge that the use of several emission scenarios would have be interesting. Nevertheless, as the moderate SSP4-4.5 scenario already sufficiently decreases the sea ice concentration to such an extent that the fast ice effect is not significant by the end of the 21st century, the use of a stronger emission scenario does not seem opportune to us to study the fast ice effect of the basal melt rate increase. Nevertheless, we added a line in the conclusion to cover that point (page 18, lines 309-312).

3. [Major] Which forcing drives the future changes in fast ice, sea ice, and ocean fields, atmospheric forcing or ocean forcing? Additional experiments to separate the effects and analyses on them are helpful for readers.

As suggested, we ran a new sets of simulations. WARM_noWind, is the same simulation as WARM but without the EC-Earth wind velocity anomalies over the atmospheric boundary conditions. WARM_noAtm, is the same simulation as WARM but without any EC-Earth3 anomaly on the atmospheric forcings. WARM_noOce, which is identical to WARM except that there are no EC-Earth3 anomaly on the ocean velocity forcing. These simulations aim to separate the ocean and the atmospheric forcing effects on the model.

As presented in Figure 1a, the EC-Earth anomaly applied at the atmospheric boundary have no effect on the ocean circulation. Indeed, WARM, WARM_noAtm and WARM_noWind have the same westward ocean transport. On the other hand, WARM_noOce shows a decreased Antarctic Slope Current (ASC) compared to WARM, which suggests that part (83%) of the intensification of the ASC in WARM compared to REF is due to the ocean velocity EC-Earth anomaly applied on the ocean boundary conditions. As the remaining part (13%) of the ASC intensification between REF and WARM is not related to the EC-Earth anomaly applied at the atmosphere, it implies that it is due to the changes in seawater density over the continental shelf. The manuscript has been adapted, with this new results and analysis (see pages 10-11, lines 177-204).
We also separate the effect of atmospheric and oceanic forcing on the changes in both sea ice and fast ice. As shown by the Figure 1b-e, the sea ice shows a sensitivity to the atmospheric velocity forcings. Without the EC-Earth3 atmospheric forcing, the sea ice concentration is higher everywhere, with a sea ice front located further north. The same effect is seen for the fast ice, with a higher fast ice frequency over the mutliyear fast ice pack without the atmospheric forcing. The manuscript has been adapted, with this new results and analysis (see page 8, lines 166-169 and Fig.A1).

[Figure]

Figure 1: Differences in ocean transport, sea ice concentration and fast ice between the WARM and WARM_noAtm simulations.

4. [Major] L110-112 I don't think that the two-year spin-up is enough to obtain the quasi-steady states in oceanic variables. In fact, large declining trends in ice-shelf basal melting are found in the first seven years (Figs. 7 and 8). Are these model drift or interannual variability? To avoid including (or decreasing) the model drift signals, results from the second cycle (after the first cycle of the 20-year run) are preferable.

To address this question, we launched a new simulation, referred to as WARM_CI. This new simulation is the same

as WARM except that it starts at the end of the WARM simulation (the equivalent of a 20yrs spin-up). As shown in Fig. 2, the ice shelf melt rates present the same drifted signals, which suggests that the drifted signal is independent of the initial conditions but is a direct consequence of the oceanic boundary conditions (here we only compare the first 10 years of simulations). The changes in basal melt rate due to the modifications of the ocean initial conditions are only observed during the first two years (more clearly in the Moscow University cavity) but then becomes negligible. All the simulations used in the manuscript have now a 20yrs spin-up (see page 5, line 122 and page 6 line 139).

[Figure]

Figure 2: Ice shelves basal melt rate for the sensitivity experiments with a full spin-up of the model (in dashed blue) for both Moscow University (top) and Totten (bottom) ice shelves. The reference simulation (in red) is the simulation presented in the paper.

5. [Major] Pelle et al. (2021) pointed out that weakening of Antarctic Slope Front/Current is important for ice-ocean interaction in this region, but the lateral boundary condition in this study is the opposite (e.g., stronger slope current in the future). It is OK there are differences among the studies. This manuscript is a numerical modeling study, and thus I suggest that the author perform additional numerical experiments to identify the role of the strength of the slope current. It is also helpful to understand the difference between the studies.

Thanks to the WARM_noOce simulation, we can compare the effect of both a low and strong ASC intensity on the warm water exchanges across the shelf. The basal melt rate in both cavities for both the WARM and WARM_noOce simulations are presented in Figure 3. For both cavities, the simulation with the higher ASC intensity (WARM)

has the lower basal melt rate. This result suggests that a stronger ASC decreases the warm water exchanges across the shelf and reduces the basal melt rate. These new results have been added to the manuscript (see page 11, lines 205-209 and Fig. A3).

[Figure]

Figure 3: Ice shelves basal melt rate for the WARM and WARM_noOce simulations for both Moscow University (top) and Totten (bottom) ice shelves. The reference simulation (in red) is the simulation presented in the paper.

6. [Major] L155-156 "This acceleration mainly results from the retreat of fast ice, ....". No evidence in the manuscript supports this sentence.
This affirmation was made based on the mean barotropic ocean velocity changes between the REF and nFST simulations. The nFST simulation shows stronger ocean current near the coast compared to REF, especially where there are packs of fast ice. The manuscript has been updated (see page 10, lines 181-182, and Fig A2 in the manuscript).

7. [Major] L161-170. To examine the ASC intensification, some analyses of the climate model (EC-Earth3) on a wider scale are required. Since the ASC is a large-scale phenomenon, not only local wind but also wind over the remote Antarctic coastal regions becomes a driving force.
We updated the conclusion to emphasise that our results on the ASC are only valid over our local domain and that the winds, through their impact on the ocean circulation outside of the configuration have an impact on oceanic forcings applied over the configuration. An EC-Earth winds analysis outside of our region of interest is beyond the scope of this research paper. Furthermore, as described above, the analysis on the separate effect of atmospheric and

oceanic forcings on the ocean circulation already provides additional information on that subject (see pages 18-19, lines 315-317).

8. [Major] L197-199 and L226-228 There are no results on sea ice production in the manuscript.
As suggested, results of sea ice production have been added to the text (see pages 9-10, lines 172-176, and figure 5).

9. [Major] I think spatial distributions related to the ice shelf/glacier basal melt rate are missing in the manuscript.
Results about the spatial distribution of the basal melt rate have been added to the manuscript (see pages 12-13, lines 228-232 and figure 8c).

**Technical corrections**

10. Figure2: Where are the locations of these observations? There are unrealistic connections in the profiles (probably connecting lines between different locations?).
The observations have been collected in two locations, in front of the Totten ice shelf cavity (these are the deeper profiles) and close to the Dalton coastal polynya (shallower profiles). The locations are now displayed in a new subfigure (see figure 2b).

11. Figures 5, 6, and 9: Please increase latitudes' tick marks (e.g., adding 65S and 67S if they are in the range).
We added more latitudes on each figures of the manuscript when possible, with ticker lines.

12. Figure3: Please use a linear scale for the vertical scale. Line or shade showing bottom topography is required for panels a-c. A vertical line showing the model domain (63S) is also helpful.
We preferred to keep the scale as it was, for better consistency with the other figures. Furthermore, since the ocean reaches more than 5000 meter depth outside the continental shelf, such a scale is useful to represent the ocean vertical profiles outside of the shelf. It keeps a large portion of the figure for the first hundred meters where most of the heterogeneity are and the rest of the figure is for the deep ocean where the ocean variables are often homogeneously distributed. Furthermore, we cut the figure at the correct domain size (the maximum latitude is now 63S). Showing the EC-Earth3 anomalies in the north of our domain was not relevant.

13. Figure 4: Please consider adding 0.75 contours in panels a-b to allow readers to compare the observational result (Fig. 1).
Done.

14. Figure 6: Please consider adding contours of the bottom topography.
Done.

15. Figures 7 and 8: Please use the same vertical scales, at least for the same regions (TIS for panel a and MUIS for panel b).
It has been done.

16. L268-269: References are required.
Done.

**Response to the Referee #2**

Dear Referee,

Thank you for the time that you spent on our manuscript. Below you will find a summary of the changes that we made throughout the manuscript to address all your suggestions.

Yours sincerely

On behalf of all the co-authors,

Guillian Van Achter

**Summary**

The Antarctic ice sheet draining into the Southern Ocean via various marine terminating glaciers - aka ice shelves is the major future contributor to global sea level rise. Melting of ice shelves is often highly influenced by the sea-ice conditions at their fronts. This study is investigating the impact of landfast sea ice in front of the Totten and Moscow University ice shelves by using a state-of-the-art coupled numerical ocean-ice model that is regionalized to the wider region of these ice shelves. The investigation focuses on the difference in the ice shelf basal melt rates between recent decades (1995-2014) and the end of the 21st century (2081-2011) - hence investigating the influence of climate warming on the environmental (atmosphere, ocean, sea ice) conditions - with and without a prognostic fast ice coverage. The main outcomes of the study are i) presence of landfast sea ice increases melting rates for both ice shelves under current conditions, ii) climate warming triggers enhanced melting rates at the Totten but not the Moscow University Ice shelves, and iii) without landfast ice the increase in melting rates due to climate warming is larger than with landfast ice.

I rate this as an appropriately well written study of a very interesting aspect. While the presentation of the figures and the material is mostly very clear, I have the impression - independent of what I wrote in my comments further below - that the manuscript would benefit from a careful reading and perhaps restructuring of the content of one or the other paragraph. One example is the one in lines 195-204. However, overall things seems sufficiently clear to me mostly. I have three general comments and only few specific and editoral comments.

**General comments**

GC1: The paper would benefit from an improved description of the physical processes that the authors expect to resolve with their study. While most of these come at a certain point in the description of the results and/or in the discussion, the readability of the paper as a whole would be greatly enhanced if the authors could come up with research hypotheses ... perhaps along the lines:

Climate warming leads to a reduction of the sea ice cover in the Southern Ocean and hence most likely to a reduction in the stability and duration of the landfast ice cover.

A reduction in landfast sea ice changes the atmosphere-ocean energy fluxes and can impact near-surface ocean currents and the vertical water mass structure.

We thank the referee for the suggestion. We improved the research hypotheses paragraph in the introduction with the proposed sentences (see page 3, lines 62-68).

GC2: There is more in the data than the authors show and discuss. This begins with the differences in the standard deviations shown in Table 2 (why?), continues with little discussion of the temporal variability inherent in the time series of the melt rates ($-->$ What happens in years 6 and 7?), and ends when it comes to incorporating observational datasets to enhance the credibility of some of the statements made - be it with respect to the design of the experiment (keyword ice bergs) or with respect to how realistic is the fast ice cover modeled / where are main ice production sites located. As suggested we added more analysis on the changes in melt rate std (see pages 12, lines 232-236, Fig. A4). The melt rate decrease of the sixth and seventh years is intrinsic to the ocean boundary conditions (see page 12, lines 227-228). The validation of the REF simulation against observation (fast ice, sea ice production, polynya locations, ..) has already been done in the Van Achter et al. (2022) paper, and we think that it would be redundant to do it again in this manuscript. Furthermore, the focus of this study is more on the comparison between REF and WARM. We adapted the manuscript to emphasise this point in the experimental design (see page 5, lines 122-124).

GC3: Some of the points discussed would benefit from more illustrative figures - such as results obtained with nFST and nFST_WARM in the context of the winter sea ice concentration (and polynya location) or the near-surface ocean currents. As suggested in some of the specific comments, we have added figures in the updated version of the manuscript (sea ice production, spatial distribution of the melt rate, ocean velocities for nFST,...)

**Specific comments**

L25-31: In these lines you refer to the effect of fast ice. While you partly differentiate between multiyear fast ice (L25) and seasonal fast ice (L30) it remains unclear whether there is difference in the impact of these two kinds. Would it make sense to be more clear here? The impact of fast ice that we have studied in our previous paper did not separate the effect of the multiyear fast ice from the seasonal fast ice. The multiyear fast ice, being located along the coast and being thicker, has an important role as insulator during the Summer, but both yearly and multiyear fast ice are important during Winter, by decreasing the sea ice production and enhancing the ocean stratification near the coast. We adapted the manuscript, see page 2, lines 31-35.

In addition I am wondering whether it would make sense at this stage, to provide more details about the physical processes by which fast ice protects an ice shelf and/or changes water mass modification such that it has a notable impact on the development of the ice shelf. Describing these processes upfront would also help to understand whether and how the fast ice in the model leads to changes in the ice shelf; are the processes the same? How does a fast ice cover change the water mass properties? How does a fast ice cover protect the ice shelf boundary? As suggested, we added more details in the manuscript on how the fast ice changes the ocean stratification and how the fast ice protects the ice shelf front (see page 2, lines 31-37 and also in page 16, lines 248-252).

It seems that calving of ice bergs at the ice shelf boundary supported by the action of ocean swell is not among the processes you are taking into account. Is that correct? You could mention this here. Indeed we do not take that into account as there are no iceberg calving in the model.

L51: I guess "Those models" refers to the models referred to in L48. Still, in order to estimate the importance (or size of the knowledge gap here) of not including fast ice it might be a good idea to mention about how many models

we are talking here. None of the studies mentioned in L48 has a fast ice representation (prescribed or prognostic). To our knowledge, only a few models have a prescribed fast ice in the Antarctic, and only two have a prognostic fast ice representation (Huot et al. (2021) and us.)

L102: Remaining questions I have with respect to the model:

- Does the model allow the water to have sub-freezing temperatures (see e.g. Haumann et al. 2020)? No, it does not.

- How does the model "grow" fast ice? The fast ice formation comes firstly through the advection of sea ice which forms ice arches between icebergs and between icebergs and the coast. Once the sea ice is trapped by these ice arches, it thickens by snow accumulation and by more sea ice advection from the East. We added more information on how our model grow fast ice in the introduction (see page 2, lines 31-33).

- How does the model treat ice shelf calving and generation of ice bergs? There are no iceberg calving in the model. Which is why we don't describe this process in our introduction. The icebergs are prescribed and are static during all the simulation

- How does the model treat marine ice / platelet ice accretion underneath the ice shelf / the fast ice? The model is only ocean–sea ice coupled. So the ice shelf thickness is prescribed and stays the same throughout the simulation. There are no platelet ice accretion underneath the ice shelf, the ice melt/grow follows the ice shelf module implemented by Mathiot et al. (2017) (temperature and velocity dependent). The fast ice is treated as sea ice in the model.

Figure 4: In the caption you (correctly) write "sea ice concentration" whereas in the title of the panels your write "sea ice extent". This should be harmonized towards "sea ice concentration" or "sea ice area fraction". Done.

Figure 5: In order to avoid readers trying to find the eastward transport associated with the ACC in panels a) and b) it might make sense to annotate more latitudes. Agreed, done.

Please remind the reader your motivation to choose a transect (in panel d) that is at the far eastern boundary of your region of interest and therefore quite far away from both the gyre on the shelf and the TIS. There was an error in the manuscript, the ocean transport is averaged over the all configuration.

L184: "more variable (+55%)" −− > It is not clear to what you are referring to here? To the increase in the standard deviation? It is now specified in the manuscript that is was related to the standard deviation.

Figure 7, panel a): What happened in years 6 and 7 in TIS? Why are melt rates so similar? The basal melt rate decrease in years 6 and 7 is inherent to the ocean boundary conditions that drive a sudden temperature drop (It is now described in the manuscript in page 12, lines 223-224).

L198: "the presence of fast ice induces less sea ice production and more sea ice melt" −− > I am not sure this global statement holds. I would think that it requires to take into account whether you are dealing with seasonal or multiyear fast ice, how far away the ice production sites are from the ice shelf boundaries and how efficient these are in the context of the production of the fast ice itself. It might be very illustrative to show two panels of the kind shown in Figure 4 e) and f) which back up your notion about the change in location of polynyas (and hence areas of high ice production). We added a figure of the differences in mean sea ice production between WARM and REF to back up our statement (pages 9-10, lines 166-170, Fig. 5).

L199: The causal link between enhanced upper ocean stratification and enhanced warm water intrusion should be

made more clear. It is not immediately understandable. Perhaps it might make sense to show maps of the kind shown in Fig. 5 a), b) that illustrate the ocean currents. One of your earlier arguments was that a loss of fast ice between REF and WARM is responsible for the intensification of the Totten shelf gyre. I am wondering how this gyre looks like in nFST and nFST_WARM. From Figure 5 it is clear that during WARM there is substantially more water transport towards the TIS than during REF. As suggested before, we have detailed the introduction section about fast ice, describing the link between the changes in ocean stratification and the intrusion of warm water into the cavities. Furthermore, we added the figure A2, which shows the gyre for REF and nFST.

Table 2: What explains the switch from a lower standard deviation for 1995-2014 for the nFST cases compared to the higher standard deviation for 2081-2100 for the same cases? The higher melt rate std is explained partly by a higher mixed layer depth variability in WARM compared to REF, which should be related to the larger amplitude of the seasonal cycle of the surface air temperature (see page 13, lines 232-236).

L229/230: This might be in part triggered by the intensification of the Totten Shelf gyre, right? It might therefore make sense to come up with a number for the increase in water mass transport (in Sv) near the northeastern edge of the TIS between REF and WARM (see Fig. 5 a, b). Part of the higher melt rate in TIS compared to MUIS is indeed due to the coastal current acceleration in front of the TIS cavity. The increase of the integrated ocean transport in front of the TIS cavity is 226%. We added this results in the manuscript (see page 10, lines 182-184).

L254-256: "we were forced ... simulations" −− > I am not on your page with this statement. There is at least one data set of ice berg distribution around Antarctica that covers more than just two months in a particular year. In addition, I'd say - if you are in doubt whether this limited data set suffices - you could at least compare your modeled fast ice extents in REF with fast ice derived from either MODIS or AMSR-E/2 satellite remote sensing observations. Should - within your period of interest - substantial differences occur in the location and stability of these ice bergs then I would assume that you would discover an increasing discrepancy between your model results and the observations. I would say this is simply about getting the correct data set to look at. Alex Fraser would be one point of contact; Nihashi Oshima another one. In Van Achter et al. (2022), we compare our simulated fast ice with observed fast ice (given by Alex Fraser) over the 2001-2010 period, and the differences between simulated and observed fast ice are acceptable. The icebergs dataset was given by Rick Smith. This dataset had the advantage of being at a high resolution (less than 1km). Since the fast ice in the Totten area has a low interannual variability, we estimated that the short period covered by the icebergs dataset was enough. The goal of the L254-256 sentence is to point the difficulty to predict the icebergs distribution by the end of the 21st century and how a drastic change in iceberg distribution could strongly alter the results of the study.

**Typos / editoral remarks**

L41: "will" −− > Is this a definite change or is this rather something that could happen? Please re-phrase in case. Done.

L113: Please clarify whether Fig. 2b shows salinity profiles before or after bias correction. Done.

L138: "winds anomaly" −− > "wind anomalies" to match with "occur". Done.

L146: Would it make sense to note that this first-year fast ice is at a different location? Yes, the first-year fast ice is at a different location in WARM compared to REF. We think that this is already explained by the sentence "the multiyear fast ice cover (frequency above 0.9) in REF is replaced by first year fast ice in WARM".

L154: If we both look at the same gyre (there is only one) then this is the southern limb of the gyre that is amplified - as is even visible in the zonal transport at 66.6 deg S. Done.

L158: "eastern" −− > "eastward". Done.

L168: "mostly function" −− > "mostly a function". Done.

L169: "This" −− > "These". Done.

L185: You could add that the variability even decreases. Done.

L200: "disappears" −− > I tend to say it shrinks but it does not disappear - at least not according to Figure 4. Agreed, done.

L218: "to broader" −− > "to a broader". Done.

L226: "and a fast ice representation" −− > I suggest to stress here one more time how accurate the this fast ice representation is compared to observations ... how accurate is it? Since the comparison between simulated and observed fast ice is more the subject of the Van Achter et al. (2022) paper and is already detailed in this paper, we prefer not to add such information in the conclusion.

L231: And because there is no speed up of any currents nearby? Agreed, done.

L241: "are similar by the end of the 21st century" −− > This is valid for TIS but not for MUIS which shows a melt rate for nFST_WARM that is about 10% larger than for WARM. Especially if we see this in relation to the melt rates for TIS between REF and nFST which also differ by an order of 10%. I therefore suggest to rephrase this statement. Agreed, done.

---

## Author Response (AR2)

**Response to the Referee #1**

Dear Referee,

Thank you for the time that you spent on our manuscript. Below you will find a summary of the changes that we made throughout the manuscript to address all your suggestions.

Yours sincerely

On behalf of all the co-authors,

Guillian Van Achter

**Summary**

This is my second review for the paper "Influence of fast ice on future ice shelf melting in the Totten Glacier area, East Antarctica" by Van Achter et al. (egusphere-2022-94). I appreciate the authors for performing additional experiments (WARM_noAtm and WARM_noOce) to respond to my previous review comments. The results from these experiments are very helpful for understanding the findings in this study. However, after reading the revised manuscript, I have almost the same concerns as the previous review.

**Specific comments**

The abstract ends with the sentence, "This highlights the importance of including a representation of fast ice to simulate realistic ice shelf melt rate increase in East Antarctica under warming conditions.", and the second half of the abstract comes from the results in Table 2. Again, I think the conclusion is quite misleading. Because there is no pronounced difference in the future ice-shelf basal melting between the numerical experiments with and without the fast ice representation, and the difference in the ice-shelf basal melting is only found in the present-day condition. The manuscript in the present form gives the readers an impression that fast ice representation can control the future ice-shelf basal melting. We regret that there may have been some confusion about this sentence. Even if the melt rate sensitivity to fast ice is low in the end of the 21st century, stating that "including fast ice under warming conditions is important" is not misleading. The Totten ice shelf indeed shows a small but nevertheless significant 5% difference in melt rate with and without fast ice by the end of the 21st century. In order to avoid any further confusion, we **quantified** the sentence in both the abstract and the conclusion to make it clear that most of the sensitivity comes from the recent past but that the sensitivity is not trivial by the end of the 21st century. The abstract sentence is now: **"This basal melt rate increase sensitivity to the fast ice is explained by the strong melt rate sensitivity for present-day conditions ($\sim$ 25% difference in m/yr) and by the low melt rate sensitivity in the end of the 21st century ($\sim$ 4% difference in m/yr). Reduction of the fast ice extent in the future induces a decrease of the melt rate that partly compensates for the increase due to warming of the ocean. This highlights the importance of including a representation of fast ice to simulate realistic ice shelf melt rate increase in East Antarctica under warming conditions"** (see abstract lines 13-18 and see the conclusion lines 293-304)

L205-209 and Figure A3. I don't think that the stronger ASC in your model suppresses heat exchange across shelf breaks. In fact, Figure 7 clearly shows warm water intrusion across the shelf breaks, and furthermore, there are no

differences in ice-shelf melting after three years in Fig A3. Related to this comment, the horizontal axis of Fig. A3 should extend to 20 years to be consistent with the other figures. Indeed, the stronger ASC does not suppress the heat exchange across the shelf, but that is not what we state in the manuscript. As discussed in the manuscript, a strong ASC tends to decrease the heat exchange across the shelf compared to a weaker ASC. This has been already mentioned in several studies, for instance in Nakayama et al. (2021) and this is confirmed by our own simulations. As a direct illustration, Figure 1 clearly shows that, with the same oceanic and atmospheric warming, the simulation with a strong ASC (WARM) has a lower basal melt rate than the simulation with a weak ASC (noOce). This is true for both cavities and for the whole period of simulation. The axis of Fig. A3 has been extended in the new version of the manuscript. Moreover, regarding your point about the warm water intrusion in Figure 7, there are indeed warmer waters across the shelf in WARM compared to REF but that does not mean that the accelerated ASC has no decreasing effect on the melt rate. This only shows that the warming due to the oceanic and atmospheric forcings has a greater effect on the melt rate than the acceleration of the ASC. We rephrase the sentence as :"**As hinted by Nakayama et al. (2021), at equivalent atmospheric and oceanic warmings, the ASC modulates the heat intrusion towards the continental shelf and the TIS and MUIS cavities. The basal melt rate for both cavities in WARM and WARM_noOce (see Fig. A3) shows higher values with low ASC intensity (WARM_noOce) and lower values with high ASC intensity (WARM). This implies that, whereas the ocean and surface air temperature increase induces the intrusion of warmer water into the cavities and higher basal melt rate, the accelerated ASC limits this basal melt rate increase. However, this ASC effect is hidden in Fig. 7 by the ocean warming due to the forcings"** (see page 13 lines 211-216).

[Figure]

(a) TIS basal melt rate

(b) MUIS basal melt rate

Figure 1: Area-averaged basal melt rate of the TIS (a) and MUIS (b) for the WARM and WARM_noOce simulations. WARM_noOce is the same simulation as WARM, except that the ASC is weaker.

Although there is a phrase "..., with particular focus on the ASC changes and their origins", I couldn't find how the ASC is changed. In the previous review, I suggested some analyses in the climate model results to understand the forcing/boundary conditions for the regional model, but the authors' response was "beyond the scope of this research". How can your readers (including me) understand the ASC changes? As mentioned before, we can only investigate the acceleration of the ASC with our boundary conditions, and everything happening outside of our model domain is not in the scope of our paper. Furthermore, we answered your previous comment by running the new sensitivity simulations WARM_noOce and WARM_noAtm, which allowed us to separate the atmospheric and oceanic forcing effect on the acceleration of the ASC. We were able to show that the ASC acceleration is not directly related to the atmospheric forcing but is related at 83% to the ocean velocity forcing. These changes in ocean velocity are likely driven by changes in winds outside our domain. However, we don't have the adequate model simulations for investigating the role of winds in the ACC acceleration outside of our domain and new experiments with the global model used to obtain the boundary conditions would be required. We agree that it is an interesting question but this is one study on its own and cannot be added as an additional side analysis to our present paper. We have removed the sentence "and their origins" in the introduction and we have rewritten our conclusion by adding more information on the role of winds in the oceanic circulation and how they will change in the future. We now state:**"Finally, as the easterly wind component is projected to weaken over the next century and will significantly impact**

the Southern ocean circulation (Neme et al., 2022), our ASC change analysis should be extended to a wider scale and to other regions" (see conclusion lines 322-324).

Appendix. Although I appreciate the additional experiments, I don't think that it is good enough to add the figures without any text/explanation. I suggest the authors integrate the figures in the Appendix into the main body and carefully restructure the contents. We followed your suggestion. The figures are now included into the main discussion.

**Typos / editoral remarks**

L31-32 "In this region, the surface covered by ..." What is the surface? ocean surface? fast-ice surface? The ocean surface. This has been clarified in the manuscript.

section 2.3 It is not clear how you performed the 20-year spinup. It is not clear how long you performed for WARM_noAtm and WARM_noOce. The 20-year spin-up is made by running the 20-year simulation twice (this is now specified).

Figure 3 Please use the rounded numbers for the horizontal axis (e.g.,-66.0, -65.0 etc.). Done.

Figure 4e-f Can you plot the boundary between fast ice and sea ice in winter? Since high sea-ice production areas (coastal polynyas) are formed at the edge of fast ice/coastline/ice front, showing the fast-ice edge would be helpful in understanding the difference in sea-ice production in Fig. 5. Done.

L181-182 It seems to me that the gyre intensification originated from the stronger ASC and the enhanced transport across shelf breaks between 120-125E in Fig. 6b. Some analysis on Warm_noOce would be helpful to separate the roles of the fast ice and lateral advection. As presented in Figure 2, the southern side of the Totten gyre, which is accelerated in WARM compared to REF, shows the same ocean velocities in both nFST, WARM_noOCE and WARM. As WARM_noOce and WARM have nearly the same ocean velocities in front of Totten, this suggests that the changes in ASC does not play any role in the acceleration of this coastal current (WARM has a much stronger ASC than WARM_noOce). Moreover, as nFST presents nearly the same ocean velocities as WARM in front of Totten, this proves that the changes in fast ice cover is the main cause of the changes in ocean current in front of Totten between WARM and REF. We adapted the section on the ocean velocity discussion in the paper with the nFST and WARM_noOce mean ocean velocity to justify this more clearly.

[Figure]

Figure 2: Mean ocean barotropic velocity for the REF (a), nFST (b), WARN_noOce (c) and WARM (d) simulations, averaged over the 1995-2014 period.

L190-204 What latitude range do you use for the quantitative comparison? The range selected was -65.6 to -64.7. This is now mentioned in the text.